# MATRIX-FREE LEAST SQUARES SOLVERS:
# VALUES, GRADIENTS, AND WHAT TO DO WITH THEM

## ABSTRACT

This paper argues that the method of least squares has significant unfulfilled potential in modern machine learning, far beyond merely being a tool for fitting linear models. To release its potential, we derive custom gradients that transform the solver into a differentiable operator, like a neural network layer, enabling many diverse applications. Empirically, we demonstrate: (i) scalability by enforcing weight sparsity on a 50 million parameter model; (ii) imposing conservativeness constraints in score-based generative models; and (iii) hyperparameter tuning of Gaussian processes based on predictive performance. By doing this, our work represents the next iteration in developing differentiable linear-algebra tools and making them widely accessible to machine learning practitioners.

## 1 INTRODUCTION

The method of least squares is commonly introduced as one of the first steps into machine learning, where it is taught as a simple approach for performing linear regression (Bishop, 2006). Although its importance is recognized, it is frequently perceived as a basic tool, perhaps overshadowed by the complex non-linear models prevalent today. We argue that this misconception is a lost opportunity, and take steps towards rectifying it by providing novel tools and use cases in the present work.

For least-squares problems like $\mathbf{x}^\star := \arg\min_{\mathbf{x}} \|\mathbf{A}\mathbf{x} - \mathbf{b}\|^2 + \lambda^2 \|\mathbf{x}\|^2$ or $\mathbf{x}^\star := \arg\min_{\mathbf{x}} \|\mathbf{x}\|^2$ s.t. $\mathbf{A}\mathbf{x} = \mathbf{b}$, consider a computational abstraction `LstSq`, which takes the linear operator $\mathbf{A}$, vector $\mathbf{b}$, and a regularization weight $\lambda$ (which could be zero), and returns the least-squares solution,

$$\mathbf{x}^\star = \texttt{LstSq}(\mathbf{A}, \mathbf{b}, \lambda). \tag{1}$$

A central message of our work is that `LstSq` is not merely a solver but – if equipped with appropriate reverse-mode derivatives – a differentiable operator like a neural network layer, and that it should be used as such. One central example will be constrained optimization of a neural network (more on this later), but the method has many applications beyond this use case. More precisely, our main contributions are the following:

1. We provide an efficient JAX implementation of `LstSq`. The least-squares solutions are computed via LSMR (Fong & Saunders, 2011), which we demonstrate is superior to alternative approaches such as solving normal equations or direct methods that instantiate the matrix.

2. We derive and implement custom reverse-differentiation rules for adaptive least-squares solvers using the adjoint method. This implementation[1] has the advantage of working for all least-squares solvers, including adaptive solvers whose iteration count is unknown a priori. Our experiments show how the custom gradients are orders of magnitude faster than unrolling the solver's forward pass.

---

[1]JAX library: [redacted]

3. We revive the null-space method (Yamashita, 1980) for constrained optimization and reformulate it as a least-squares problem. This new perspective enables using our efficient implementation of `LstSq`. To the best of our knowledge, ours is the first use of the null-space method in deep learning.

4. We demonstrate the least-squares-null-space method's efficacy on a diverse set of constrained optimization tasks, such as: equivariance, weight sparsity, and conservativeness of score-based generative models, on *models with up to 50 million parameters*. We also provide a JAX library that seamlessly integrates the null-space method into Optax (DeepMind et al., 2020), allowing constrained optimization of neural networks with just a few lines of code.[2]

5. We use the backward pass of the differentiable `LstSq` solver to calibrate a Gaussian process, directly optimizing the posterior fit instead of marginal-likelihood optimization. To achieve this, we exploit the natural connection between Gaussian process regression and least squares problems, and find that targeting the posterior fit improves both runtime and quality of fit over typical baselines.

## 2 LEAST SQUARES: VALUES, GRADIENTS, AND WHAT TO DO WITH THEM

### 2.1 VALUES: MATRIX-FREE LEAST-SQUARES IN JAX

For any $m, n \in \mathbb{N}$, let $\mathbf{b} \in \mathbb{R}^m$ and $\mathbf{A} \in \mathbb{R}^{m \times n}$ be given. Throughout this work, we assume that the matrix $\mathbf{A}$ has full rank, and is parameterized by some $\theta$, $\mathbf{A} = \mathbf{A}(\theta)$. We distinguish least-squares problems with a tall $\mathbf{A}(\theta)$, which means $m \geq n$, from problems with a wide matrix, $m \leq n$. We also distinguish regularized from unregularized problems, depending on whether $\lambda$ is zero or not. Intuitively, least squares problems seek optimal solutions $\mathbf{x}$ to linear systems $\mathbf{A}\mathbf{x} = \mathbf{b}$, with a possibly-nonsquare matrix $\mathbf{A} \in \mathbb{R}^{m \times n}$ (Björck, 2024). More precisely, the least-squares method solves

$$\mathbf{x}^{\star}(\theta, \mathbf{b}, \lambda) = \texttt{LstSq}(\mathbf{A}(\theta), \mathbf{b}, \lambda) = \begin{cases} \arg\min_{\mathbf{x}} \|\mathbf{A}(\theta)\mathbf{x} - \mathbf{b}\|^2 + \lambda^2\|\mathbf{x}\|^2 & \text{if } \lambda \neq 0 \text{ or } \mathbf{A} \text{ is tall,} \\ \arg\min_{\mathbf{x}} \left\{ \|\mathbf{x}\|^2 \text{ s.t. } \mathbf{A}(\theta)\mathbf{x} = \mathbf{b} \right\} & \text{if } \lambda = 0 \text{ and } \mathbf{A} \text{ is wide.} \end{cases} \quad (2)$$

For tall $\mathbf{A}(\theta)$, the regularized problem is well-defined for $\lambda \to 0$ (in the sense of admitting a unique solution in the limit). For wide $\mathbf{A}(\theta)$, this is not the case. Instead, the relationship between the two is that the regularized problem ($\lambda \neq 0$) is solved by

$$\mathbf{x}^{\star}(\theta, \mathbf{b}, \lambda) = \mathbf{A}(\theta)^{\top}(\mathbf{A}(\theta)\mathbf{A}(\theta)^{\top} + \lambda^2\mathbb{I})^{-1}\mathbf{b} = (\mathbf{A}(\theta)^{\top}\mathbf{A}(\theta) + \lambda^2\mathbb{I})^{-1}\mathbf{A}(\theta)^{\top}\mathbf{b}. \quad (3)$$

In the limit of $\lambda \to 0$, only one of the two parameterizations in Equation 3 is well-defined, depending on whether $\mathbf{A}(\theta)$ is tall or wide. The corresponding limit of $\mathbf{x}^{\star}$ for $\lambda \to 0$ minimizes $\|\mathbf{A}\mathbf{x} - \mathbf{b}\|^2$ if $\mathbf{A}(\theta)$ is tall, or $\{\|\mathbf{x}\|^2 \text{ s.t. } \mathbf{A}(\theta)\mathbf{x} = \mathbf{b}\}$ if $\mathbf{A}(\theta)$ is wide, respectively; more on this relationship in Appendix C.

The various methods for numerically solving least squares problems can be broadly classified across two axes. Along one axis, there are *direct* versus *matrix-free* methods, which differ in whether or not the matrix $\mathbf{A}(\theta)$ is instantiated in memory ("direct method", high memory demands, and for realistic problems, $\mathbf{A}(\theta)$ is too big to store in memory) or only accessed via matrix-vector products ("matrix-free methods", low memory demands). Along the other axis, there are approaches based on solving the linear system in Equation 3 versus

Table 1: **Approaches to solving least squares problems.** "L.S.": linear system; "O.T.": orthogonal transformation.

| Method | Property | Direct | Matrix-free |
|--------|----------|--------|-------------|
| **L.S.** | Examples | Cholesky | Conj. Grad. |
| | Precision | Double | Double |
| | Memory | $\mathcal{O}(\min(m, n)^2)$ | $\mathcal{O}(\max\{m, n\})$ |
| **O.T.** | Examples | SVD, QR | LSMR |
| | Precision | Single | Single |
| | Memory | $\mathcal{O}(mn)$ | $\mathcal{O}(\max\{m, n\})$ |

---

[2]JAX library: `[redacted]`

those approaches that apply orthogonal transformations to $\mathbf{A}(\theta)$ and extract $\mathbf{x}^\star$ directly, for example, us-ing Golub-Kahan-Li bidiagonalization (Golub & Kahan, 1965). Methods that solve linear systems require twice the precision of methods that orthogonally transform $\mathbf{A}(\theta)$. This is because the solvers compute $\mathbf{v} \mapsto (\mathbf{A}(\theta)^\top \mathbf{A}(\theta) + \lambda^2 \mathbb{I})^{-1} \mathbf{v}$, which means that singular values of $\mathbf{A}(\theta)$ are squared, affecting the condition-ing of the problem accordingly. Table 1 summarises the differences in approaches. Only matrix-free methods that target orthogonal transformations combine low memory demands with the ability to work in low precision, which is why we focus on this class of solvers in the present work. As an instance of such an algorithm, our implementation uses LSMR (Fong & Saunders, 2011). LSMR is equivalent to applying MINRES (Paige & Saunders, 1975a) to the linear system in Equation 3, but more robust because it handles matrix-vector and vector-matrix products with $\mathbf{A}(\theta)$ separately, not in combination (thus it avoids "squaring"; Appendix A). However, there are also scenarios where direct methods, which target orthogonal transformations like QR decomposition, can be advantageous, especially in cases where the matrix in question is small and can be stored in memory.

Commonly, for example, in SciPy's implementation (Virtanen et al., 2020), LSMR requires access to both $\mathbf{v} \mapsto \mathbf{A}(\theta)\mathbf{v}$ and $\mathbf{u} \mapsto \mathbf{A}(\theta)^\top \mathbf{u}$. Transposing the linear operator $\mathbf{A}(\theta)$ manually, without instantiating $\mathbf{A}(\theta)$, is often tedious and a common source of error. To avoid this error source, our framework handles transposition automatically: Accessing only $\mathbf{v} \mapsto \mathbf{A}(\theta)\mathbf{v}$, the transposed linear operator emerges from automatic differentiation. A vector-Jacobian product (for instance, using `jax.vjp` or `torch.func.vjp`),

$$\left[ \mathbf{A}(\theta)\mathbf{v}_0, \left( \mathbf{u} \mapsto \mathbf{A}(\theta)^\top \mathbf{u} \right) \right] = \text{vjp} \left( \mathbf{v} \mapsto \mathbf{A}(\theta)\mathbf{v}, \mathbf{v}_0 \right) \tag{4}$$

yields both the value $\mathbf{A}(\theta)\mathbf{v}$ and a function that implements matrix-vector products with $\mathbf{A}(\theta)^\top$; see also Potapczynski et al. (2023). At every step of LSMR's forward pass, we call this vector-Jacobian product and thereby transpose $\mathbf{A}(\theta)$ without exposing the possibility of erroneous implementations of transpose linear operators. Furthermore, this automatic transposition only requires a single backward pass through a function that is known to be linear, which is very efficient (Radul et al., 2023) . By reducing sources of errors, the solvers become more practical, which is important for using numerical least squares in modern machine-learning toolchains.

## 2.2 Gradients: Adjoint of the least-squares problem

Assume that `LstSq` accesses the matrix $\mathbf{A}$ only through parameterized matrix-vector products, which means $(\theta, \mathbf{v}) \mapsto \mathbf{A}(\theta)\mathbf{v}$. If the solution $\mathbf{x}^\star = \mathbf{x}^\star(\theta, \mathbf{b}, \lambda)$ of the least-squares problem (Equation 3) is then passed to a downstream loss function $\mu = \mu(\mathbf{x}^\star)$, we need a backward pass (think, "gradient") through `LstSq` to optimize $\mu$ with respect to $\theta$, $\mathbf{b}$, or $\lambda$. We never optimize $\mu$ with respect to $\mathbf{A}$, only with respect to $\theta$, because if $\mathbf{A}$ is too big to store in memory, $\nabla_\mathbf{A} \mu$ would be as well. The central challenge tackled next is the computation of the gradients of this overall loss $\mu$ with respect to the underlying parameters $\theta$, $\mathbf{b}$, and $\lambda$ – that is, computing $\nabla_\theta \mu$, $\nabla_\mathbf{b} \mu$, and $\nabla_\lambda \mu$ from $\nabla_\mathbf{x} \mu$. These gradients then enable end-to-end differentiation of computational pipelines featuring least-squares problems. The following theorem states how to implement this backward pass, and is an essential contribution of this work.

**Theorem 1** (Gradients of `LstSq`). *Let $\mathbf{A}(\theta)$ be a full-rank matrix, dependent on parameters $\theta$, and accessed through matrix-vector products $(\theta, \mathbf{v}) \mapsto \mathbf{A}(\theta)\mathbf{v}$. Let $\mathbf{b}$ be a known vector, and let $\lambda \in \mathbb{R}$ be a known regularization weight. Let $\mu$ be a scalar objective function that depends on the solution of a least-squares problem involving $\mathbf{A}(\theta)$, $\mathbf{b}$, and $\lambda$. Then, the following two statements hold for any $\lambda \in \mathbb{R}$:*

*1. Suppose $\mathbf{x}^\star$ solves the least-squares problem in Equation 2 with a **tall matrix** $\mathbf{A}(\theta)$, then, we have*

$$\nabla_\theta \mu = \nabla_\theta g(\theta), \quad \nabla_\mathbf{b} \mu = \text{LstSq}(\mathbf{A}(\theta)^\top, \nabla_\mathbf{x} \mu, \lambda), \quad \nabla_\lambda \mu = 2\lambda \langle \xi, \mathbf{x} \rangle, \tag{5}$$

*with $g(\theta) := \langle \mathbf{r}, \mathbf{A}(\theta)\xi \rangle + \langle \nabla_\mathbf{b} \mu, \mathbf{A}(\theta)\mathbf{x}^\star \rangle$, $\mathbf{r} := \mathbf{A}(\theta)\mathbf{x}^\star - \mathbf{b}$, and $\xi := \text{LstSq}(\mathbf{A}(\theta), \nabla_\mathbf{b} \mu, 0)$.*

2. *Suppose* $\mathbf{x}^\star$ *solves the least-squares problem in Equation 2 with a* **wide matrix** $\mathbf{A}(\theta)$, *then, we have*

$$\nabla_\theta\mu = \nabla_\theta g(\theta), \quad \nabla_\mathbf{b}\mu = \mathtt{LstSq}(\mathbf{A}(\theta)^\top, \nabla_\mathbf{x}\mu, \lambda), \quad \nabla_\lambda\mu = -2\lambda\langle\nabla_\mathbf{b}\mu, \mathbf{y}\rangle, \quad (6)$$

*with* $g(\theta) \coloneqq \langle\nabla_\mathbf{b}\mu, \mathbf{A}(\theta)\mathbf{x}\rangle + \langle\mathbf{y}, \mathbf{A}(\theta)\mathbf{r}\rangle$, $\mathbf{r} \coloneqq \mathbf{A}(\theta)^\top\nabla_\mathbf{b}\mu - \nabla_\mathbf{x}\mu$ *and* $\mathbf{y} \coloneqq \mathtt{LstSq}(\mathbf{A}(\theta)^\top, \mathbf{x}, 0)$.

*Proof.* The essential strategy for deriving these gradient expressions is to use the method of adjoints. An introduction to the latter is in Appendix B. A complete proof of the theorem can be found in Appendix C. □

**Related work on Theorem 1:** To the best of our knowledge, Theorem 1 is new. However, similar-but-different statements have been made in prior work. The results most closely related to Theorem 1 are those by Golub & Pereyra (1973) and Krämer et al. (2024). Golub & Pereyra (1973) derive forward-mode derivatives of pseudo-inverses, which are closely linked with least-squares solvers. In contrast, Theorem 1 states the reverse-mode derivatives, and handles a regularisation term. A gradient with respect to this term will be needed in the experiment in Section 3.2. Krämer et al. (2024) derive efficient recursions for backward passes through Lanczos and Arnoldi methods, and use them to compute gradients of matrix functions. Conversely, our work derives backward passes through numerical least-squares solvers, albeit using similar proof techniques. Finally, Amos & Kolter (2017)'s work on implicit layers shares a high-level theme with our work, but the technical contributions (numerical least squares versus quadratic programs) and applications differ entirely. Blondel et al. (2022) implements software for implicit differentiation of various optimality conditions for general problems. Our work differs from this by being more specialised and focusing on least-squares problems, and this specialisation gives us quite a significant advantage in computational efficiency (essentially by avoiding general-purpose linear system solvers like CG or GMRES). CoLA (Potapczynski et al., 2023) focuses on exploiting compositional matrix structure for scalable linear algebra scale applied to modern machine learning scale problems. Our work is complementary: CoLA provides efficient forward and generic automatic differenitation rules for structured operators, while we show that for least-squares problems, one can go further by designing custom adjoints, yielding particularly efficient differentiation of least-squares solvers.

## 2.3 WHAT TO DO WITH THEM: CONSTRAINED OPTIMIZATION OF NEURAL NETWORKS

In the remainder of this paper, we turn to novel applications of least squares in deep learning. Specifically, we focus on constrained optimization of neural networks. Numerous desirable properties, such as physical principles, equivariance, or sparsity, can be incorporated through model constraints. Consider the problem

$$\theta^\star = \arg\min_{\theta\in\mathbb{R}^d}\left\{\mathbb{E}_{x\sim\mathcal{X}}\left[\mathcal{L}(\theta, x)\right] \text{ s.t. } \mathbf{c}(\theta) = \mathbf{0}\right\}, \quad (7)$$

where $\theta \in \mathbb{R}^d$ represents the network parameters, $\mathcal{L}$ is the task loss, and $\mathbf{c} : \mathbb{R}^d \to \mathbb{R}^k$ defines $k \in \mathbb{N}$ constraint, which shall be continuously differentiable and we assume that the number of constraints are smaller than parameter dimension. The loss and $\theta$ are unrelated to those in previous sections. Table 6 (Appendix) shows examples for constraints appearing in combination with neural networks. Standard optimizers like Adam (Kingma & Ba, 2015) are ill-suited for solving Equation 7, since they operate exclusively in the unconstrained parameter space. However, the solution $\theta^\star$ of Equation 7 must satisfy the Karush-Kuhn-Tucker conditions (Nocedal & Wright, 1999): primal feasibility (satisfying the constraint) and Lagrangian stationarity. Finding a $\theta^\star$ that simultaneously satisfies both conditions can be challenging, particularly for non-linear constraints in high-dimensional parameter spaces encountered in deep learning.

The *null-space method* (Yamashita, 1980) proposes an iterative algorithm that circumvents these issues. Instead of enforcing the constraint directly, Yamashita (1980) proposes to enforce its first-order approximation, which amounts to solving a sequence of local problems with linearized constraints; for some $\theta_t \in \mathbb{R}^d$, it enforces

$$\mathbf{c}(\theta) \approx \mathbf{c}(\theta_t) + \mathbf{J}_\mathbf{c}(\theta_t)(\theta - \theta_t) = \mathbf{0}. \quad (8)$$

```
import optax
from nuox import linalg, nsm  # null-space method

def constraint(params):  # example constraint: enforce unit norm
    return jnp.dot(params, params) - 1.0

transform = nsm.projection(constraint, solver=linalg.lsmr())
optim = optax.chain(transform,  optax.adam(1e-3))  # Use any optax optimizer
```

Figure 1: Combine the null-space projection with a standard Optax optimizer using `redacted`.

An algorithm is derived by studying a differential equation whose critical points are the solution to an equality-constrained optimization problem. Discretizing such a flow with learning rates $\eta, \gamma > 0$ yields

$$\theta_{t+1} = \theta_t - \eta \left( \mathbb{I} - \mathbf{J_c}(\theta_t)^\top (\mathbf{J_c}(\theta_t)\mathbf{J_c}(\theta_t)^\top)^{-1}\mathbf{J_c}(\theta_t) \right) \nabla \mathcal{L}(\theta_t) + \gamma \mathbf{J_c}(\theta_t)^\top (\mathbf{J_c}(\theta_t)\mathbf{J_c}(\theta_t)^\top)^{-1}\mathbf{c}(\theta_t). \quad (9)$$

We make the crucial observation that this update can be reformulated as a least-squares problem:

$$\theta_{t+1} - \theta_t = -\arg\min_\delta \left\{ \frac{1}{2}||\delta - \eta\nabla_\theta\mathcal{L}(\theta_t))||^2 \text{ s.t. } \mathbf{J_c}(\theta_t)\delta = -\gamma\mathbf{c}(\theta_t) \right\} \quad (10a)$$

$$= -\eta\nabla_\theta\mathcal{L}(\theta_t) + \texttt{LstSq}\big(\mathbf{J_c}(\theta_t), \eta\mathbf{J_c}(\theta_t)\nabla_\theta\mathcal{L}(\theta_t) - \gamma\mathbf{c}(\theta_t), 0\big). \quad (10b)$$

Appendix D explains how to use `LstSq` for a least-squares problem with a bias term (in other words, how to transition from Equation 10a to Equation 10b). Crucially, *the transformation of the gradient in Equation 10b turns any unconstrained optimizer into one for the constrained problem in Equation 7.* This is beneficial because state-of-the-art stochastic optimization routines can now be used for solving constrained optimization problems. We exploit this generality of the null-space method in our code implementation: Figure 1 shows that with our library, a few lines of code can turn any of Optax's (DeepMind et al., 2020) gradient transformations into an algorithm for solving constrained optimization problems.

Yamashita (1980) shows that under appropriate assumptions, the null-space method converges to the solution of the constrained optimization problem and that it has a quadratic rate of convergence. To make such results accessible to a more general audience, Appendix E *provides a new proof* of convergence. As for a geometric interpretation, constrained optimization can be thought of as optimization on a manifold (Boumal, 2023).

Using this perspective, null-space-method steps can then be derived as Riemannian gradient steps with the projection onto the tangent space as approximations of the exponential map. Appendix F elaborates on this *new interpretation of the null-space method using differential geometry*.

Classical methods for constrained optimization include penalty methods, Lagrangian methods, and projected gradient descent (Nocedal & Wright, 1999). Penalty methods turn a constrained problem into an unconstrained one by adding a penalty term to the loss. However, for finite penalty weights, penalty methods do

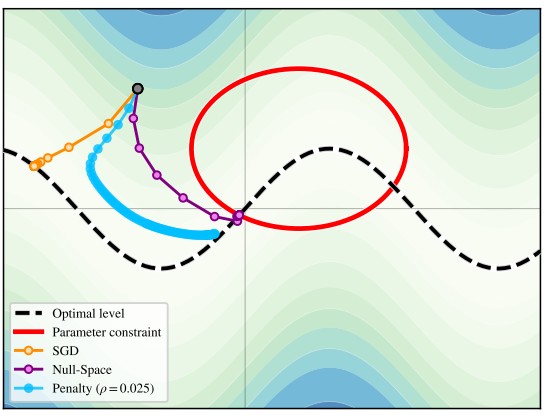

Figure 2: SGD & penalty method fail the constraint, unlike the null-space method.

not satisfy primal feasibility upon convergence (Nocedal & Wright, 1999), and a large penalty weight can distort the optimization landscape, leading to poor solutions for the task loss. Figure 2 shows this issue.

Another approach to constrained optimization is to optimize an (augmented) Lagrangian. However, this requires finding saddle points rather than minima, which complicates the use of typical gradient-based (stochastic) optimization routines, leading to intricate update schemes that can be difficult to tune (Walsh, 1975). Finally, projected gradient descent (PGD) is an iterative method that alternates between a standard gradient update on the task loss and a projection step onto the feasible set defined by $\mathbf{c}(\theta) = \mathbf{0}$. While conceptually straightforward, the projection step is computationally expensive or analytically intractable for most constraints. Consequently, PGD is often limited to simple problems, and does not generalize to the applications we demonstrate in Section 3.

Table 2 summarizes the relative strengths and weaknesses of different classical methods.

Recent works that tackle constrained optimization for neural networks have combined one of the aforementioned classical methods with certain approximations. Gallego-Posada et al. (2022) find a saddle point of the Lagrangian by doing gradient descent on the neural network parameters and gradient ascent on the Lagrange multiplier. Donti et al. (2021) propose a framework for constrained optimization, which is equivalent to an approximate projected gradient descent scheme. Our usage of the null-space method is the first of its kind in a deep learning setting, and generally novel in combination with numerical least squares.

Table 2: Key properties of various constrained optimization algorithms. "NSM":"Null-space method". PGD: "Projected gradient descent."

|  | NSM | Penalty | Lagr. | PGD |
|---|---|---|---|---|
| KKT | ✓ | ✗ | ✓ | ✓ |
| No saddle pts. | ✓ | ✓ | ✗ | ✓ |
| Any constraint | ✓ | ✓ | ✓ | ✗ |

## 3 EXPERIMENTS

### 3.1 EFFICIENCY OF CUSTOM GRADIENTS

Next, we compare the efficiency of our custom gradient (Theorem 1) with automatic differentiation "through" an adaptive least-squares solver (LSMR), implemented via Equinox's reverse-mode differentiable while-loops (Kidger & Garcia, 2021). As a test problem, let $\mathbf{A}$ be a square convolution matrix with a fixed-size convolution kernel and an increasing number of rows and columns. The convolution kernel as well as the right-hand side vector $\mathbf{b}$ are randomly sampled from $\mathcal{N}(\mathbf{0}, \mathbb{I})$. We measure the runtime (wall time), reporting the fastest of three runs to minimize "machine noise" as much as possible – the results are in Figure 3. The runtimes of all three are proportional, but our custom backward pass is five to ten times faster than the alternatives.

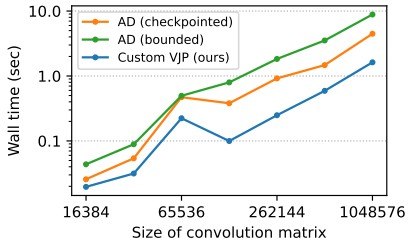

Figure 3: Automatic differentiation versus a custom vector-Jacobian product (VJP). Our custom VJP is five to ten times faster than unrolling the solver's loop.

### 3.2 GAUSSIAN PROCESS CALIBRATION VIA DIFFERENTIABLE LEAST-SQUARES

Next, we demonstrate the utility of reverse derivatives of adaptive least-squares codes. Gaussian processes are a natural testbed for two reasons: first, they are closely linked to least-squares problems (Williams & Rasmussen, 2006); second, matrix-free linear algebra is popular for accelerating Gaussian process inference (Gardner et al., 2018). Consider the task of interpolating a function $f_{\text{true}}(x) = \cos(2\pi x) + x \sin(5\pi x)$ from noisy observations. We sample 1600 training data points $\mathbf{X}_{\text{train}}$, and 400 test data points $\mathbf{X}_{\text{test}}$ uniformly on $[0, 1]$, with corresponding noisy evaluations $\mathbf{y}_{\text{train}}$ and $\mathbf{y}_{\text{test}}$. The noise is discussed below. For unknown

lengthscale $\ell$, output-scale $\sigma$, and observation noise $\sigma$, consider the following probabilistic model. Let $\omega \sim \mathcal{N}(\mathbf{0}, \mathbb{I}_k/\ell^2)$, and $b \sim \mathcal{U}([0, 2\pi])$ be fixed and define a Gaussian process with $k = 200$ random Fourier features (Rahimi & Recht, 2007),

$$\mathbf{z} \sim \mathcal{N}(\mathbf{0}, \mathbb{I}_k), \quad f(x) = \Phi_{\sigma,\ell}(x)\mathbf{z} = \left[\sigma \cos(\omega_i^\top x + b)z_i\right]_{i=1}^k, \quad \mathbf{y} \mid \mathbf{z} \sim \mathcal{N}(f(\mathbf{X})\mathbf{z}, \lambda^2 \mathbb{I}) \quad (11)$$

where $(\mathbf{X}, \mathbf{y})$ represent the in- and outputs of either the training or the test set, respectively, depending on the stage of the experiment. Equation 11 models isotropic Gaussian observation noise, but we generate the data with anisotropic noise (increasing as $x$ increases; see Figure 4). This model mismatch emulates what is typically encountered when using Gaussian processes "in the wild". Given the training data, the conditional mean $\mathbf{z}^\star := \mathbb{E}[\mathbf{z} \mid \mathbf{y}_{\text{train}}]$ solves a least-squares problem,

$$\mathbf{z}^\star(\sigma, \ell, \lambda) = \arg\min_{\mathbf{z}} \left\{ \|\Phi_{\sigma,\ell}(\mathbf{X}_{\text{train}})\mathbf{z} - \mathbf{y}_{\text{train}}\|^2 + \lambda^2 \|\mathbf{z}\|^2 \right\} = \texttt{LstSq}\left(\Phi_{\sigma,\ell}(\mathbf{X}_{\text{train}}), \mathbf{y}_{\text{train}}, \lambda\right). \quad (12)$$

We compute the solution to this least-squares problem using LSMR, selecting the tolerance $10^{-5}$. Then, we learn the hyperparameters $\ell$, $\sigma$, and $\lambda$ using two different algorithms:

1. **Baseline:** Type-II log-marginal-likelihood optimization on the training set (Williams & Rasmussen, 2006), which minimizes the negative log-probability-density function of the observations $\mathbf{y}_{\text{train}}$

$$L(\sigma, \ell, \lambda) := -\log p(\mathbf{y}_{\text{train}} \mid \sigma, \ell, \lambda) = -\log \mathcal{N}(\mathbf{y}_{\text{train}} \mid 0, \Phi_{\sigma,\ell}(\mathbf{X}_{\text{train}})\Phi_{\sigma,\ell}(\mathbf{X}_{\text{train}})^\top + \lambda^2 \mathbb{I}) \quad (13)$$

   to find the optimal hyperparameters. Type-II marginal likelihood optimisation is the typical calibration strategy for Gaussian process models (Williams & Rasmussen, 2006) and, thus, the baseline.

2. **Ours:** Evaluating the fit of the predictive mean, which means that we first compute $\mathbf{z}^\star(\sigma, \ell, \lambda)$ according to Equation 12, and then evaluate

$$L(\sigma, \ell, \lambda) := \|\Phi_{\sigma,\ell}(\mathbf{X}_{\text{train}})\mathbf{z}^\star(\sigma, \ell, \lambda) - \mathbf{y}\|^2 + \lambda^2 \|\mathbf{z}^\star(\sigma, \ell, \lambda)\|. \quad (14)$$

   This approach is surprisingly uncommon in the Gaussian process literature – we only know of Nguyen et al. (2021) who use it – but beats marginal likelihood in simplicity and scalability (shown below). Evaluating the gradient of this calibration loss requires gradients of $\texttt{LstSq}$.

Both losses are optimized with standard optimizers and learning rates. The results of this comparison for 10 different random seeds are in Figure 4. They show how using our predictive-mean calibration loss is more than ten times faster (left plot), with lower test loss (root-mean-square error on test data, middle plot; the $p$-value is 3.11%, which suggests that the differences are significant), and a better visual fit (right). The mean-data fit shows how the marginal-likelihood strategy leads to underfitting in four of the ten cases, whereas our loss consistently performs well. In summary, the differentiable LSMR code enables highly efficient calibration of Gaussian process models.

### 3.3 CONSTRAINED OPTIMIZATION

Next, we demonstrate the capabilities of the null-space method using various practical constraints. The focal points are, next to good performance, versatility, and ease of application; thus, the benchmarks below prefer baselines that are typical to each constraint over those that are carefully-tuned state-of-the-art implementations. To make things fair, we use equally little fine-tuning for our approach – the results are surprisingly strong. We anticipate that domain-specific optimizations could further enhance performance and scalability in each application. Precise setups for all experiments are in Appendix H.

**Enforcing equivariance:** The null-space method enables the enforcement of complex functional properties like equivariance and invariance directly during training, without the need for bespoke architectures. These properties act as powerful structural priors, guiding the model to learn representations that respect known symmetries or are robust to specific nuisance transformations in the input data.

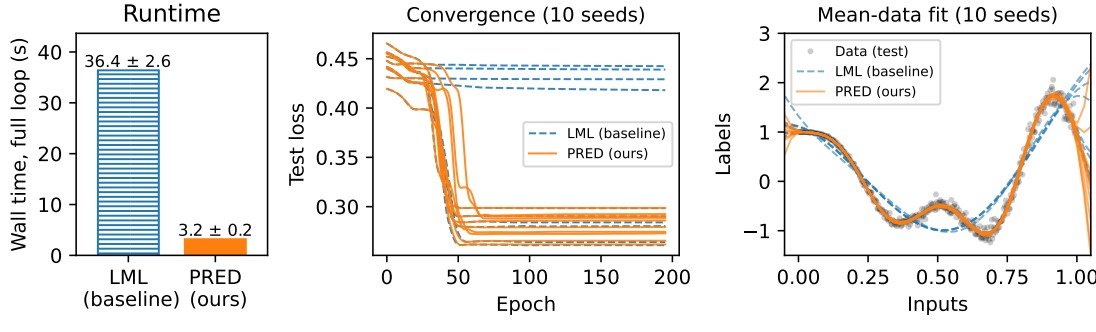

Figure 4: Calibration with negative log-marginal-likelihood (LML, baseline) versus evaluating the fit of the predictive mean (PRED, ours). PRED is over ten times faster (left), consistently achieves lower test losses (root mean square error on test set, $p$-value is $3.11\%$), which correlates with a better mean-data fit (right).

Table 3: Comparing the null-space method with baselines on $C_4$ and $O(3)$ equivariance. Test error (accuracy and mean-square error) and constraint violation. Lower is better.

| Structure | Task | Method | Test Error ($\downarrow$) | Constraint Violation ($\downarrow$) |
|---|---|---|---|---|
| $C_4$ Equiv. | FMNIST | Baseline | $\mathbf{0.105 \pm 0.004}$ | $743.51 \pm 141.65$ |
| $C_4$ Equiv. | FMNIST | Null-space (ours) | $0.147 \pm 0.004$ | $\mathbf{0.27 \pm 0.12}$ |
| $O(3)$ Equiv. | From (Finzi et al., 2021) | Data augm. (baseline) | $0.13 \pm 0.01$ | $0.36 \pm 0.01$ |
| $O(3)$ Equiv. | From (Finzi et al., 2021) | Null-space (ours) | $\mathbf{0.11 \pm 0.01}$ | $\mathbf{0.18 \pm 0.01}$ |

- *Rotation Equivariance:* We enforce $C_4$ rotational equivariance (Cohen & Welling, 2016) on the convolutional layers of a LeNet (Lecun et al., 1998) model trained on FMNIST (Xiao et al., 2017). The constraint $\mathbf{c}(\theta)$ minimizes the difference between final feature maps resulting from rotating the input image and rotating the final output feature maps, ensuring the learned filters respect $C_4$ rotational symmetry (i.e., $f(R\mathbf{x}; \theta) - Rf(\mathbf{x}; \theta) = \mathbf{0}$ for $C_4$ rotation $R$). The results in Table 3 show that the null-space method balances the task loss and constraint satisfaction: it achieves slightly lower test accuracy compared to "ordinarily trained" models but exhibits vastly superior satisfaction of $C_4$ equivariance.

- $O(3)$ *Equivariance:* A key strength of our null-space method is its ability to impose complex group equivariances by changing just a few lines of code to define the constraint. We demonstrate this by enforcing $O(3)$ equivariance on the task of predicting the moment of inertia for particle systems (Finzi et al., 2021). We randomly sample transformations $R \in O(3)$ and the constraint $f(R\mathbf{x}; \theta) - R^\top f(\mathbf{x}; \theta)R = \mathbf{0}$. Then, we benchmark our null-space method against a baseline that uses $O(3)$ data augmentation. Performance is evaluated on a test dataset by measuring the mean squared error (MSE) on a test set and the $O(3)$ equivariance constraint violation (Table 3). The null-space method outperforms data augmentation in both metrics.

**Enforcing $\ell_0$ sparsity:** Enforcing a specific $\ell_0$ sparsity level during training can be achieved using learnable stochastic masks (Louizos et al., 2018; Gallego-Posada et al., 2022). We optimize mask-probabilities $\mathbf{p}$ alongside model weights $\theta$, and sample masks from a Bernoulli distribution, using Yin et al. (2019)'s straight-through estimator for $\mathbf{p}$'s gradients derived from the task loss. Our constrained optimization method is applied

Table 4: Test accuracy (%) for models trained to target $\ell_0$ sparsity. Higher is better.

| Model | Dataset | Method | Test accuracy (%, ↑) |
|-------|---------|--------|----------------------|
| ResNet-18 | CIFAR-10 (90% Sparsity) | Magnitude pruning (baseline) | $0.6830 \pm 0.0139$ |
|           |                         | Constrained $\ell_0$ (ours) | $\mathbf{0.7825 \pm 0.0015}$ |
| ResNet-18 | SVHN (90% Sparsity) | Magnitude pruning (baseline) | $0.7987 \pm 0.0273$ |
|           |                     | Constrained $\ell_0$ (ours) | $\mathbf{0.9150 \pm 0.0025}$ |
| SWIN-S | ImageNet (50% Sparsity) | Magnitude pruning (baseline) | $0.468$ |
|        |                         | Constrained $\ell_0$ (ours) | $\mathbf{0.498}$ |

Table 5: Comparing the null-space method, USBMs, and QCSBM on three datasets: 8-Gaussian, Spirals, and Checkerboard. NLL is measured in bits/dimension. Lower is better.

| Dataset/Model | Asym (↓) | NAsym (↓) | Score Error (↓) | NLL (↓) |
|---------------|----------|-----------|-----------------|---------|
| **8-Gaussian** | | | | |
| Null-Space | $\mathbf{2.66 \pm 1.34} \cdot 10^{-3}$ | $\mathbf{3.20 \pm 1.47} \cdot 10^{-4}$ | $1.52 \pm 0.08$ | $\mathbf{3.70 \pm 0.03}$ |
| USBM | $2.38 \pm 0.25 \cdot 10^{-2}$ | $3.74 \pm 0.09 \cdot 10^{-3}$ | $1.50 \pm 0.06$ | $3.79 \pm 0.09$ |
| QCSBM | $6.96 \pm 1.20 \cdot 10^{-3}$ | $1.39 \pm 0.06 \cdot 10^{-3}$ | $\mathbf{1.49 \pm 0.05}$ | $3.74 \pm 0.07$ |
| **Spirals** | | | | |
| Null-Space | $\mathbf{3.37 \pm 0.10} \cdot 10^{-3}$ | $\mathbf{1.21 \pm 0.07} \cdot 10^{-3}$ | $1.63 \pm 0.08$ | $\mathbf{3.53 \pm 0.15}$ |
| USBM | $6.68 \pm 3.48 \cdot 10^{-1}$ | $3.43 \pm 0.78 \cdot 10^{-2}$ | $1.57 \pm 0.07$ | $4.11 \pm 0.01$ |
| QCSBM | $5.13 \pm 1.55 \cdot 10^{-2}$ | $9.43 \pm 0.44 \cdot 10^{-3}$ | $\mathbf{1.53 \pm 0.04}$ | $4.02 \pm 0.05$ |
| **Checkerboard** | | | | |
| Null-Space | $\mathbf{4.26 \pm 2.35} \cdot 10^{-3}$ | $\mathbf{9.87 \pm 5.21} \cdot 10^{-4}$ | $1.65 \pm 0.09$ | $\mathbf{3.69 \pm 0.05}$ |
| USBM | $9.15 \pm 1.10 \cdot 10^{-2}$ | $1.91 \pm 0.16 \cdot 10^{-2}$ | $1.65 \pm 0.09$ | $3.74 \pm 0.07$ |
| QCSBM | $2.17 \pm 0.26 \cdot 10^{-2}$ | $5.86 \pm 0.52 \cdot 10^{-3}$ | $\mathbf{1.64 \pm 0.04}$ | $3.76 \pm 0.01$ |

to $\mathbf{p}$ via a constraint $c(\mathbf{p}) = \frac{1}{N_p} \sum_{i=1}^{N_p} p_i - s_{\text{target}} = 0$, where $N_p$ is the total number of parameters, which drives the expected proportion of active weights towards a target sparsity level $s_{\text{target}}$. We apply this method to train ResNet-18 (He et al., 2016) on CIFAR-10 (Krizhevsky, 2009) (target $s_{\text{target}} = 0.1$, i.e., 90% sparsity) and SWIN-S (Liu et al., 2021) on ImageNet (Deng et al., 2009; Russakovsky et al., 2015) (target $s_{\text{target}} = 0.5$, i.e., 50% sparsity). We compare against one-shot magnitude pruning (Lee et al., 2024) as a baseline. Table 4 shows that the null-space method achieves better test accuracies than magnitude pruning.

**Conservative property of score-based generative models:** Score-based generative models learn the score function $\mathbf{s}(\mathbf{x}; \theta) = \nabla_{\mathbf{x}} \log p_d(\mathbf{x})$, where $p_d(\mathbf{x})$ is the data distribution. For $\mathbf{s}$ to be a valid score function, it must be a conservative vector field, implying its Jacobian must be symmetric, $\mathbf{J}(\mathbf{s}) - \mathbf{J}(\mathbf{s})^\top = \mathbf{0}$; see (Chao et al., 2023) for details. We apply the null-space method to enforce this conservativeness constraint $\mathbf{c}(\theta) = ||\mathbf{J}(\mathbf{s}) - \mathbf{J}(\mathbf{s})^\top||_F^2 = 0$ during training. This encourages score-based models that are both architecturally flexible and theoretically sound. We compare our null-space method to typical unconstrained score-based models (USBMs) and Chao et al. (2023)'s quasi-conservative score-based models (QCSBM) on various synthetic 2D datasets. The performance is evaluated via asymmetry (Asym), normalized asymmetry (NAsym), score error, and negative log-likelihood (NLL). The results in Table 5 show that the null-space method outperforms the baselines by achieving the best NLL and a stricter enforcement of conservativeness.

The experiment code is under `[redacted]`.

## 4 LIMITATIONS AND CONCLUSION

This paper is the first step towards rectifying the misconception that least squares is a basic tool and only useful for linear regression. To this end, our work explains how to compute values and (novel) gradients of matrix-free least squares solvers, offering JAX code that seamlessly embeds the now-differentiable `LstSq` operator into modern deep learning software stacks.

The first main contribution of this article was the backward pass through `LstSq` (Theorem 1), which requires exactly two extra forward passes per gradient. However, while efficient, our gradient expressions are currently limited to full-rank matrices, and future work should investigate the case of rank-deficient systems.

The second main contribution is the revitalization of the null-space method, an algorithm by Yamashita (1980) that relies heavily on numerical least squares via gradient projections. Our implementation of the null-space method is not just effective, as demonstrated on a range of experiments (Section 3), but it's also incredibly simple: all experiments use the same few lines of JAX code (Figure 1). However, the null-space method incurs an additional computational overhead on top of the underlying gradient-based optimization, which is the cost of solving the least-squares problem at each update step. Fortunately, the computational complexity of our least-squares solver of choice (LSMR) is linear in the number of rows and columns, and its space complexity matches that of standard gradient-based optimizers. And, unlike in Yamashita (1980)'s article, our experiments always mini-batch the data to account for deep-learning-sized datasets. While empirically, this choice proved effective, future work should analyze the convergence of such a stochastic variant of the null-space method. In any case, the constrained optimization of neural networks has become considerably easier, which means that many exciting applications can now be built on top of these advancements.

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

## A  LEAST SQUARES REDUX

At its core, a least-squares problem seeks an optimal solution $\mathbf{x}$ to a linear system $\mathbf{Ax} = \mathbf{b}$, where $\mathbf{A} \in \mathbb{R}^{m \times n}$ and $\mathbf{b} \in \mathbb{R}^m$ (Björck, 2024). The precise formulation depends on the dimensions of $\mathbf{A}$. If $m \geq n$, $\mathbf{A}$ is a tall matrix, and the least-squares problem is about finding $\mathbf{x} \in \mathbb{R}^n$ that minimizes the squared Euclidean norm of the residual, $\arg\min_{\mathbf{x}} ||\mathbf{Ax} - \mathbf{b}||^2$. The solution of the tall least-squares problem is the pseudo-inverse, which simplifies to $\mathbf{x}^\star = (\mathbf{A}^\top \mathbf{A})^{-1} \mathbf{A}^\top \mathbf{b}$ if $\mathbf{A}$ has full column rank. If $m \leq n$, $\mathbf{A}$ is a wide matrix and the goal is to find $\arg\min_{\mathbf{x}} \frac{1}{2} ||\mathbf{x}||^2$ subject to $\mathbf{Ax} = \mathbf{b}$. Like in the tall case, the solution is the pseudo-inverse. If $\mathbf{A}$ has full row rank, it simplifies to $\mathbf{x}^\star = \mathbf{A}^\top (\mathbf{A}\mathbf{A}^\top)^{-1} \mathbf{b}$. Throughout this paper, we assume $\mathbf{A}$ is too large for explicit instantiation, accessed only via matrix-vector and vector-matrix products ("matvecs", "vecmats") $\mathbf{v} \mapsto \mathbf{Av}$.

Several numerical strategies exist for solving the least-squares problem, each with implications for efficiency and stability depending on the problem's structure. One common strategy for solving the least-squares problem (illustrated with the wide case) is via the *normal equations*. This involves solving $(\mathbf{A}\mathbf{A}^\top)\mathbf{y}^\star = \mathbf{b}$ for $\mathbf{y}^\star$, followed by $\mathbf{x}^\star = \mathbf{A}^\top \mathbf{y}^\star$. For smaller $\mathbf{m}$ (number of rows in $\mathbf{A}$), $\mathbf{A}\mathbf{A}^\top$ can be formed explicitly and solved with direct methods like Cholesky factorizations. For larger $m$ where forming $\mathbf{A}\mathbf{A}^\top$ is infeasible, for example if $\mathbf{A}$ is the Jacobian of a neural network, iterative matrix-free solvers such as the conjugate gradient (CG, Hestenes et al., 1952) or minimum residual method (MINRES, Paige & Saunders, 1975b) can be applied, requiring only matrix-vector products with $\mathbf{A}$ and $\mathbf{A}^\top$. However, methods relying on normal equations suffer a critical drawback: squaring the matrix $\mathbf{A}$ exacerbates ill-conditioning (the eigenvalues of $\mathbf{A}\mathbf{A}^\top$ are the squared singular values of $\mathbf{A}$), leading to numerical instability and slow convergence for iterative solvers. An example follows shortly.

As an alternative to solving normal equations, bidiagonalization methods offer a numerically robust foundation for large-scale least-squares problems, particularly when $\mathbf{A}$ is accessed only via matrix-vector products. The

standard algorithm for this is the Golub-Kahan iterative bidiagonalization (Golub & Kahan, 1965). This iterative process, after $k$ iterations and with starting vector $\mathbf{b}$, generates two matrices with orthonormal columns, $\mathbf{U} \in \mathbb{R}^{m \times k}$ and $\mathbf{V} \in \mathbb{R}^{n \times k}$, and a lower bidiagonal matrix $\mathbf{B} \in \mathbb{R}^{k \times k}$. These matrices are such that $\mathbf{A} \approx \mathbf{U} \mathbf{B} \mathbf{V}^\top$ and $\mathbf{V} \mathbf{e}_1 = \mathbf{b}/\|\mathbf{b}\|$ holds, where $\mathbf{e}_1$ is the first unit basis vector. The quality of this approximation depends on the singular values of $\mathbf{A}$; details are in the book by Golub & Van Loan (2013). Conceptually, if the process were run for enough iterations (e.g., $k = \min(m,n)$ assuming full rank), it would yield a full factorization $\mathbf{A} = \mathbf{U} \mathbf{B} \mathbf{V}^\top$, but the process is rarely run for that long. The approximation $\mathbf{A} \approx \mathbf{U} \mathbf{B} \mathbf{V}^\top$, $\mathbf{V} \mathbf{e}_1 = \mathbf{b}/\|\mathbf{b}\|$ yields

$$\mathbf{A}(\mathbf{A}^\top \mathbf{A})^{-1}\mathbf{b} \approx \mathbf{U}\mathbf{B}\mathbf{V}^\top(\mathbf{V}\mathbf{B}^\top\mathbf{U}^\top\mathbf{U}\mathbf{B}\mathbf{V}^\top)^{-1}\mathbf{b} = \|\mathbf{b}\|\mathbf{U}(\mathbf{B}^\top)^{-1}\mathbf{e}_1. \tag{15}$$

Since only a linear system involving $\mathbf{B}^\top$ needs to be solved, squaring of matrices is circumvented. This results in significantly improved numerical stability and often more rapid convergence to an accurate solution; see Example 2.

**Example 2** (Bidiagonalization vs. CG). *Consider the following least-squares problem: A is a randomly populated $10^5 \times 50$ matrix with singular values in $[1, 1/\epsilon]$ where $\epsilon$ is machine precision ($\approx 10^{-7}$). Least-squares based on bidiagonalisation is closely related to solving the normal equations with CG (Paige & Saunders, 1982), but the fact that solving the normal equation via CG requires $\mathbf{A}\mathbf{A}^\top$, whereas bidiagonalization handles $\mathbf{A}$, affects the numerical reliability of the algorithm; see Figure 5. For well-conditioned matrices, the choice between CG and bidiagonalization would not matter much. But for ill-conditioned matrices, where numerical robustness is important, solving least squares problems with bidiagonalization instead of CG is vital.*

The bidiagonalization solver from Equation 15 is more robust and efficient than solving the normal equations with CG, but could still be improved: Equation 15 requires access to $\mathbf{U} \in \mathbb{R}^{m \times k}$, storing which is prohibitive for large problems. There exist error-adaptive, $\mathcal{O}(\max\{m, n\})$)-memory versions of bidiagonalization solvers, namely, LSQR and LSMR (Paige & Saunders, 1982; Fong & Saunders, 2011), which avoid storing $\mathbf{U}$ or $\mathbf{V}$. LSMR and LSQR are mathematically equivalent to applying MIN-RES, respectively, CG to the normal equations (Paige & Saunders, 1982; Fong & Saunders, 2011), but are more robust because they use bidiagonalization. In the remainder of this article, when we discuss least-squares solution operators, $\mathbf{x}^\star = \texttt{LstSq}(\mathbf{A}, \mathbf{b}, \lambda)$, we mean LSMR, unless specified otherwise. JAX code for LSMR is provided under the URL in the main paper.

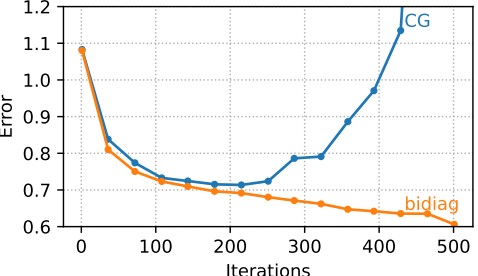

Figure 5: Bidiagonalization vs. CG.

## B BACKGROUND ON THE METHOD OF ADJOINTS

The method of adjoints offers a powerful technique for computing gradients of an objective function, say $\mu$, with respect to parameters, say $\theta$, and "through" an algorithm $\theta \mapsto \mathbf{x}$ whose outputs are implicitly defined by a set of constraints, $\mathbf{f}(\theta, \mathbf{x}) = \mathbf{0}$. The general procedure involves four key steps (Krämer et al., 2024). (i) Identify the constraint that the algorithm's inputs and outputs must satisfy. (ii) Differentiate the constraints to obtain a linear relationship between the differentials. This typically takes the form

$$\frac{\partial \mathbf{f}(\theta, \mathbf{x})}{\partial \mathbf{x}}\mathbf{dx} + \frac{\partial \mathbf{f}(\theta, \mathbf{x})}{\partial \theta}\mathbf{d}\theta = \mathbf{0}, \tag{16}$$

where $\frac{\partial \mathbf{f}(\theta, \mathbf{x})}{\partial \mathbf{x}}$ and $\frac{\partial \mathbf{f}(\theta, \mathbf{x})}{\partial \theta}$ are the Jacobians of the constraint, and $d\mu$, $d\mathbf{x}$, and $d\theta$ are infinitesimal perturbations. (iii) Introduce an adjoint variable (respectively Lagrange multiplier) $\xi$, combine it with Equation 16, and add

it to the differential $d\mu = \langle \nabla_{\mathbf{x}}\mu, d\mathbf{x} \rangle$. This leads to an expression:

$$d\mu = \langle \nabla_{\mathbf{x}}\mu, \mathbf{dx} \rangle + \left\langle \xi, \frac{\partial \mathbf{f}(\theta, \mathbf{x})}{\partial \mathbf{x}} \mathbf{dx} + \frac{\partial \mathbf{f}(\theta, \mathbf{x})}{\partial \theta} \mathbf{d}\theta \right\rangle \tag{17a}$$

$$= \left\langle \nabla_{\mathbf{x}}\mu + \left( \frac{\partial \mathbf{f}(\theta, \mathbf{x})}{\partial \mathbf{x}} \right)^{\top} \xi, \mathbf{dx} \right\rangle + \left\langle \left( \frac{\partial \mathbf{f}(\theta, \mathbf{x})}{\partial \theta} \right)^{\top} \xi, \mathbf{d}\theta \right\rangle. \tag{17b}$$

(iv) To find the gradient $\nabla_{\theta}\mu$, identify the adjoint system $\nabla_{\mathbf{x}}\mu + \left( \frac{\partial \mathbf{f}(\theta, \mathbf{x})}{\partial \mathbf{x}} \right)^{\top} \xi = \mathbf{0}$. Solving for $\xi$ leads to $\nabla_{\theta}\mu = \left( \frac{\partial \mathbf{f}(\theta, \mathbf{x})}{\partial \theta} \right)^{\top} \xi$. Details: (Krämer et al., 2024; Blondel & Roulet, 2024). The advantage of the adjoint method over other forms of deriving reverse-mode derivatives of computer programs is that it only requires an inner product and a constraint. Thus, it not only applies to vector-valued problems, but also to matrix- or function-space valued algorithms without much modification. Therefore, we use the adjoint method for deriving reverse-mode derivatives of least squares codes.

## C  PROOF OF THEOREM 1

To prove Theorem 1, we apply the four steps involved in the method of adjoints (Appendix B) to the regularized least-squares problem defined in Equation 2. We distinguish the cases of tall versus wide $\mathbf{A}$, because their behaviours are slightly different in the limit of the regulariser approaching zero.

Let $\mathbf{A}(\theta)$, $\mathbf{b}$, and $\lambda$ be known. Assume that $\mathbf{A}$ has full rank. Recall the least-squares objective

$$\mathcal{L}(\mathbf{x}) = \|\mathbf{A}(\theta)\mathbf{x} - \mathbf{b}\|^2 + \lambda^2 \|\mathbf{x}\|^2 \tag{18}$$

with minimum $\mathbf{x}^\star = \arg\min_{\mathbf{x}} \mathcal{L}(\mathbf{x})$. Any such least-squares solution $\mathbf{x}^\star$ satisfies $\nabla_{\mathbf{x}}\mathcal{L} = \mathbf{0}$, which means

$$\mathbf{A}(\theta)^{\top}(\mathbf{A}(\theta)\mathbf{x}^\star - \mathbf{b}) + \lambda^2 \mathbf{x}^\star = \mathbf{0}. \tag{19}$$

Reorder the terms to obtain

$$\mathbf{x}^\star = (\mathbf{A}(\theta)^{\top}\mathbf{A}(\theta) + \lambda^2\mathbb{I})^{-1}\mathbf{A}(\theta)^{\top}\mathbf{b} = \texttt{LstSq}(\mathbf{A}(\theta), \mathbf{b}, \lambda). \tag{20}$$

This defines the $\texttt{LstSq}$ operator; however, in practice, we do not evaluate $\texttt{LstSq}$ with Equation 20, but with our implementation of LSMR. Due to the "push-through identity' $(\mathbf{A}(\theta)^{\top}\mathbf{A}(\theta) + \lambda^2\mathbb{I})^{-1}\mathbf{A}(\theta)^{\top} = \mathbf{A}(\theta)^{\top}(\mathbf{A}(\theta)\mathbf{A}(\theta)^{\top} + \lambda^2\mathbb{I})^{-1}$ holds, there is an equivalent representation of $\mathbf{x}^\star$,

$$\mathbf{x}^\star = \mathbf{A}(\theta)^{\top}(\mathbf{A}(\theta)\mathbf{A}(\theta)^{\top} + \lambda^2\mathbb{I})^{-1}\mathbf{b}. \tag{21}$$

Both representations in Equation 20 and Equation 21 are the same, but they behave differently for $\lambda \to 0$ and for different shapes of $\mathbf{A}(\theta)$. If $\mathbf{A}(\theta)$ is tall (and has full rank), then $\mathbf{A}(\theta)^{\top}\mathbf{A}(\theta)$ is invertible but $\mathbf{A}(\theta)\mathbf{A}(\theta)^{\top}$ is not; and vice versa for when $\mathbf{A}(\theta)$ is wide. Since we want the gradients to hold for both wide and tall matrices, and for any $\lambda$, we distinguish tall and wide settings below.

### C.1  TALL CASE

In the remainder of this section, we refer to $\mathbf{x}^\star$ as $\mathbf{x}$ and to $\mathbf{A}(\theta)$ as $\mathbf{A}$. Assume $\mathbf{A}$ has full rank. If $\mathbf{A}$ is tall, we apply the adjoint method to the constraint

$$(\mathbf{A}^{\top}\mathbf{A} + \lambda^2\mathbb{I})\mathbf{x} = \mathbf{A}^{\top}\mathbf{b} \tag{22}$$

because this expression always yields a unique $\mathbf{x}$ even for $\lambda \to 0$. Differentiate the constraint,

$$\mathbf{dA}^{\top}\mathbf{Ax} + \mathbf{A}^{\top}\mathbf{dAx} + \mathbf{A}^{\top}\mathbf{Adx} + 2\lambda d\lambda \mathbf{x} + \lambda^2 \mathbf{dx} = \mathbf{dA}^{\top}\mathbf{b} + \mathbf{A}^{\top}\mathbf{db}. \tag{23}$$

For any scalar $\mu = \mu(\mathbf{x})$, let $\xi$ be any vector with as many dimensions as there are constraints. Let $\nabla_{\mathbf{x}}\mu$ be given. Then,

$$d\mu = \langle \nabla_{\mathbf{x}}\mu, \mathbf{dx} \rangle \tag{24a}$$

$$= \langle \nabla_{\mathbf{x}}\mu, \mathbf{dx} \rangle + \langle \xi, \mathbf{dA}^\top \mathbf{Ax} + \mathbf{A}^\top \mathbf{dAx} + \mathbf{A}^\top \mathbf{A}\mathbf{dx} + 2\lambda d\lambda \mathbf{x} + \lambda^2 \mathbf{dx} - \mathbf{dA}^\top \mathbf{b} - \mathbf{A}^\top \mathbf{db} \rangle \tag{24b}$$

$$= \langle \mathbf{z_x}, \mathbf{dx} \rangle + \langle \mathbf{z_b}, \mathbf{db} \rangle + \langle \mathbf{Z_A}, \mathbf{dA} \rangle + \langle z_\lambda, d\lambda \rangle \tag{24c}$$

for variables

$$\mathbf{z_x} := \nabla_{\mathbf{x}}\mu + (\mathbf{A}^\top \mathbf{A} + \lambda^2 \mathbb{I})\xi \tag{25a}$$

$$\mathbf{z_b} := -\mathbf{A}\xi \tag{25b}$$

$$\mathbf{Z_A} := \mathbf{Ax}\xi^\top + \mathbf{A}\xi\mathbf{x}^\top - \mathbf{b}\xi^\top \tag{25c}$$

$$z_\lambda := 2\lambda \langle \xi, \mathbf{x} \rangle. \tag{25d}$$

Due to the rules of the adjoint method, if $\mathbf{z_x} = 0$, then $\mathbf{z_b} = \nabla_{\mathbf{b}}\mu$, $\mathbf{Z_A} = \nabla_{\mathbf{A}}\mu$, and $z_\lambda = \nabla_\lambda\mu$ hold. Thus,

$$\nabla_{\mathbf{b}}\mu = \mathbf{A}(\mathbf{A}^\top \mathbf{A} + \lambda^2 \mathbb{I})^{-1}\nabla_{\mathbf{x}}\mu = \texttt{LstSq}(\mathbf{A}, \nabla_{\mathbf{x}}\mu, \lambda) \tag{26a}$$

$$\xi = -(\mathbf{A}^\top \mathbf{A} + \lambda^2 \mathbb{I})^{-1}\nabla_{\mathbf{x}}\mu = (\mathbf{A}^\top \mathbf{A})^{-1}\mathbf{A}^\top \nabla_{\mathbf{b}}\mu = \texttt{LstSq}(\mathbf{A}^\top, \nabla_{\mathbf{b}}\mu, 0) \tag{26b}$$

$$\nabla_{\mathbf{A}}\mu = (\mathbf{Ax} - \mathbf{b})\xi^\top + (\nabla_{\mathbf{b}}\mu)\mathbf{x}^\top \tag{26c}$$

$$\nabla_\lambda\mu = 2\lambda \langle \xi, \mathbf{x} \rangle \tag{26d}$$

are the desired gradients. We can evaluate the required quantities with the same least-squares solver that has been employed in the forward pass.

If $\mathbf{A}$ depends on parameters $\theta$, then $\nabla_{\mathbf{A}}\mu$ can be turned into $\nabla_\theta\mu$ via (abbreviate $\mathbf{r} := \mathbf{Ax} - \mathbf{b}$),

$$d\mu = \langle \nabla_{\mathbf{A}(\theta)}\mu, \mathbf{dA}(\theta) \rangle + \text{const} \tag{27a}$$

$$= \left\langle \mathbf{r}\xi^\top + (\nabla_{\mathbf{b}}\mu)\mathbf{x}^\top, \frac{\partial}{\partial\theta}\mathbf{A}(\theta)d\theta \right\rangle + \text{const} \tag{27b}$$

$$= \nabla_\theta g(\theta) + \text{const}, \tag{27c}$$

where "+const" means that the line depends on quantities that are not related to $\mathbf{dA}(\theta)$, and where $g$ is defined as $g(\theta) := \langle \mathbf{r}, \mathbf{A}(\theta)\xi \rangle + \langle \nabla_{\mathbf{b}}\mu, \mathbf{A}(\theta)\mathbf{x} \rangle$. Once $\mathbf{r}$, $\xi$, $\mathbf{x}$, and $\nabla_{\mathbf{b}}$ are available, $g(\theta)$ and its $\theta$-gradient can be evaluated with automatic differentiation. This concludes the gradients for the tall case.

## C.2 WIDE CASE

For the wide case, we proceed in the same way but we apply the adjoint method to the constraints

$$\mathbf{x} = \mathbf{A}^\top \mathbf{y}, \qquad (\mathbf{AA}^\top + \lambda^2 \mathbb{I})\mathbf{y} = \mathbf{b} \tag{28}$$

because these imply a well-defined $\mathbf{x}$ if $\mathbf{A}$ is wide, even for $\lambda \to 0$ (we always assume $\mathbf{A}$ is full rank). Differentiate the constraints

$$\mathbf{dx} = \mathbf{dA}^\top \mathbf{y} + \mathbf{A}^\top \mathbf{dy}, \tag{29a}$$

$$\mathbf{dAA}^\top \mathbf{y} + \mathbf{AdA}^\top \mathbf{y} + \mathbf{AA}^\top \mathbf{dy} + 2\lambda d\lambda \mathbf{y} + \lambda^2 \mathbf{dy} = \mathbf{db} \tag{29b}$$

Let $\mu = \mu(\mathbf{x})$ be a scalar function. Let $\mathbf{p}$ and $\mathbf{q}$ be two vectors with the same dimensions as the two constraints. Then,

$$d\mu = \langle \nabla_{\mathbf{x}}\mu, \mathbf{dx} \rangle \tag{30a}$$

$$= \langle \nabla_{\mathbf{x}}\mu, \mathbf{dx} \rangle + \langle \mathbf{p}, -\mathbf{dx} + \mathbf{dA}^\top \mathbf{y} + \mathbf{A}^\top \mathbf{dy} \rangle \tag{30b}$$

$$+ \langle \mathbf{q}, -\mathbf{dAA}^\top \mathbf{y} + \mathbf{AdA}^\top \mathbf{y} + \mathbf{AA}^\top \mathbf{dy} + 2\lambda d\lambda \mathbf{y} + \lambda^2 \mathbf{dy} - \mathbf{db} \rangle \tag{30c}$$

$$= \langle \mathbf{z_x}, \mathbf{dx} \rangle + \langle \mathbf{z_y}, \mathbf{dy} \rangle + \langle \mathbf{z_b}, \mathbf{db} \rangle + \langle \mathbf{Z_A}, \mathbf{dA} \rangle + \langle z_\lambda, d\lambda \rangle \tag{30d}$$

with the variables

$$\mathbf{z_x} = \nabla_\mathbf{x}\mu - \mathbf{p} \tag{31a}$$

$$\mathbf{z_y} = \mathbf{Ap} + \mathbf{AA}^\top\mathbf{q} + \lambda^2\mathbf{q} \tag{31b}$$

$$\mathbf{z_b} = -\mathbf{q} \tag{31c}$$

$$\mathbf{Z_A} = \mathbf{yp}^\top - \mathbf{qy}^\top\mathbf{A} + \mathbf{yq}^\top\mathbf{A} = \mathbf{yp}^\top - \mathbf{qx}^\top + \mathbf{yq}^\top\mathbf{A} \tag{31d}$$

$$z_\lambda = 2\lambda\langle\mathbf{q}, \mathbf{y}\rangle \tag{31e}$$

If $\mathbf{z_x} = 0$ and $\mathbf{z_y} = 0$, then by the adjoint method, $\mathbf{z_b} = \nabla_\mathbf{b}\mu$, $\mathbf{Z_A} = \nabla_\mathbf{A}\mu$, and $z_\lambda = \nabla_\lambda\mu$ holds.

Before solving $\mathbf{z_x} = 0$ and $\mathbf{z_y} = 0$ for suitable $\mathbf{p}$ and $\mathbf{q}$, note how we can obtain $\mathbf{y}$ from the least-squares solution $\mathbf{x}$ with another least-squares call, since

$$\mathbf{y} = (\mathbf{AA}^\top + \lambda^2\mathbb{I})^{-1}\mathbf{b} = (\mathbf{AA}^\top)^{-1}\mathbf{Ax} = \mathtt{LstSq}(\mathbf{A}^\top, \mathbf{x}, 0) \tag{32}$$

holds. Now, $\mathbf{z_x} = 0$ implies $\mathbf{p} = \nabla_\mathbf{x}\mu$, and thus

$$\mathbf{q} = (\mathbf{AA}^\top + \lambda^2\mathbb{I})^{-1}\mathbf{A}\nabla_\mathbf{x}\mu = \mathtt{LstSq}(\mathbf{A}, \nabla_\mathbf{x}\mu, \lambda) \tag{33}$$

is another least-squares call. Therefore,

$$\nabla_\mathbf{b}\mu = \mathtt{LstSq}(\mathbf{A}, \nabla_\mathbf{x}\mu, \lambda) \tag{34}$$

as well as

$$\nabla_\mathbf{A}\mu = \mathbf{yr}^\top + \nabla_\mathbf{b}\mu\mathbf{x}^\top, \qquad \mathbf{r} = \mathbf{A}^\top\nabla_\mathbf{b}\mu - \nabla_\mathbf{x}\mu, \qquad \nabla_\lambda\mu = -2\lambda\langle\nabla_\mathbf{b}\mu, \mathbf{y}\rangle. \tag{35}$$

If $\mathbf{A}$ depends on $\theta$, like in the tall case, we can turn $\nabla_\mathbf{A}\mu$ into $\nabla_\theta\mu$ by.

$$d\mu = \langle\nabla_\mathbf{A}\mu, \mathbf{dA}\rangle + \text{const} = \left\langle\nabla_\mathbf{b}\mu\mathbf{x}^\top + \mathbf{yr}^\top, \frac{\partial}{\partial\theta}\mathbf{A}d\theta\right\rangle + \text{const} = \nabla_\theta g(\theta) + \text{const}, \tag{36}$$

where $g(\theta) = \langle\nabla_\mathbf{b}\mu, \mathbf{Ax}\rangle + \langle\mathbf{y}, \mathbf{Ar}\rangle$. As soon as $\mathbf{y}$, $\mathbf{r}$, $\mathbf{x}$, and $\nabla_\mathbf{b}\mu$ are available, $\nabla_\theta g$ can be evaluated with automatic differentiation. This concludes the proof.

## D  WEIGHTED LEAST-SQUARES

In this section, we show how it is no loss of generality to consider unweighted least-squares problems only. We show how we can use an unweighted least-squares code to solve weighted problems. We only discuss the wide case, because in the tall case, absorbing the weights and biases in $\mathbf{A}$ and $\mathbf{b}$ is relatively straightforward.

Specifically, consider the weighted least-squares problem:

$$\arg\min\|\mathbf{Wx} - \mathbf{v}\|^2 \quad \text{subject to} \quad \mathbf{Ax} = \mathbf{b}. \tag{37}$$

Here, $\mathbf{W}$ is tall or square, $\mathbf{A}$ is wide or square, and $\mathbf{v}$ and $\mathbf{b}$ are vectors. Substitute $\mathbf{z} := \mathbf{Wx} - \mathbf{v}$:

$$\arg\min\|\mathbf{z}\|^2 \quad \text{subject to} \quad \begin{cases} \mathbf{Ax} = \mathbf{b} \\ \mathbf{z} = \mathbf{Wx} - \mathbf{v} \end{cases}. \tag{38}$$

Reorganise $\mathbf{z} = \mathbf{Wx} - \mathbf{v}$ into $\mathbf{x} = \mathbf{W}^+(\mathbf{z} + \mathbf{v})$, with pseudoinverse $\mathbf{W}^+$ (same shape as $\mathbf{W}^\top$):

$$\arg\min\|\mathbf{z}\|^2 \quad \text{subject to} \quad \mathbf{AW}^+\mathbf{z} = \mathbf{b} - \mathbf{AW}^+\mathbf{v}. \tag{39}$$

Solve for $\mathbf{z}$ with standard least-squares code. Then, get $\mathbf{x}$ via $\mathbf{x} = \mathbf{W}^+(\mathbf{z} + \mathbf{v})$.

# E CONVERGENCE OF THE NULL-SPACE METHOD

**Theorem 3.** *Define a continuously differentiable constraint function $\mathbf{c} : \mathbb{R}^D \to \mathbb{R}^O$, where $D$ is the number of neural network parameters and $O$ is the number of constraints. We assume $O \leq D$. If the update rule in Equation 44 converges to a point $\theta^*$, then the point satisfies:*

1. *Primal Feasibility: The constraint is satisfied, i.e., $\mathbf{c}(\theta^*) = 0$.*

2. *Lagrangian Stationarity: There exists some $\lambda^*$ such that $\nabla \mathcal{L}(\theta^*) = \mathbf{J_c}(\theta^*)^\top \lambda^*$.*

*Proof.* The null-space update step can be written explicitly as (recall $O \leq D$):

$$\theta_{t+1} - \theta_t = -\eta \arg\min_\delta \left\{ \frac{1}{2}||\delta - \eta\nabla_\theta\mathcal{L}(\theta_t))||^2 \quad \text{s.t.} \quad \mathbf{J_c}(\theta_t)\delta = -\gamma\mathbf{c}(\theta_t) \right\} \tag{40a}$$

$$= -\eta\nabla_\theta\mathcal{L}(\theta_t) - \mathtt{LstSq}\big[\mathbf{J_c}(\theta_t), \mathbf{J_c}(\theta_t)\eta\nabla_\theta\mathcal{L}(\theta_t) - \gamma\mathbf{c}(\theta_t)\big] \tag{40b}$$

$$= -\eta\big[(\mathbb{I} - (\mathbf{J_c}(\theta_t))^+\mathbf{J_c}(\theta_t))\nabla_\theta\mathcal{L}(\theta_t)\big] - \gamma\big[(\mathbf{J_c}(\theta_t))^+\mathbf{c}(\theta_t)\big], \tag{40c}$$

$$\text{where} \quad (\mathbf{J_c}(\theta_t))^+ := \mathbf{J_c}(\theta_t)^\top \left(\mathbf{J_c}(\theta_t)\mathbf{J_c}(\theta_t)^\top\right)^{-1}. \tag{40d}$$

Here, $(\mathbf{J_c}(\theta_t))^+$ is the pseudo-inverse, which means it satisfies $\mathbf{J_c}(\theta_t)(\mathbf{J_c}(\theta_t))^+ = \mathbb{I}$.

We begin by proving primal feasibility. We observe that:

$$\mathbf{J_c}(\theta_k)(\theta_{k+1} - \theta_k) = -\mathbf{J_c}(\theta_k)\big[\eta\left(\mathbb{I} - (\mathbf{J_c}(\theta_t))^+\mathbf{J_c}(\theta_t)\right)\nabla_\theta\mathcal{L}(\theta_t) + \gamma(\mathbf{J_c}(\theta_t))^+\mathbf{c}(\theta_t)\big] \tag{41a}$$

$$= -\eta\left(\mathbf{J_c}(\theta_k) - \mathbf{J_c}(\theta_k)\right)\nabla_\theta\mathcal{L}(\theta_t) - \gamma\mathbf{J_c}(\theta_t)(\mathbf{J_c}(\theta_t))^+\mathbf{c}(\theta_t) \tag{41b}$$

$$= -\gamma\,\mathbf{c}(\theta_t). \tag{41c}$$

Since we assume that the update rule converges to some $\theta^\star$, we have that $\theta_{k+1} - \theta_k \to \mathbf{0}$, this implies that as $k \to \infty$

$$\mathbf{c}(\theta^\star) = -\frac{1}{\gamma}\mathbf{J_c}(\theta_k)(\theta_{k+1} - \theta_k) \longrightarrow -\frac{1}{\gamma}\mathbf{J_c}(\theta^\star)\mathbf{0} = \mathbf{0}. \tag{42}$$

Therefore $\mathbf{c}(\theta^\star) = \mathbf{0}$. This gives us primal feasibility.

To show Lagrangian stationarity, we observe that the updates and the constraint value are both $\mathbf{0}$ at convergence. This implies that Equation 40c approaches zero which in the limit, of $k \to \infty$, leads to:

$$\nabla_\theta\mathcal{L}(\theta^\star) = \mathbf{J_c}(\theta^\star)^\top \left(\mathbf{J_c}(\theta^\star)\mathbf{J_c}(\theta^\star)^\top\right)^{-1}\mathbf{J_c}(\theta^\star)\nabla_\theta\mathcal{L}(\theta^\star). \tag{43}$$

Define $\lambda^\star = \left(\mathbf{J_c}(\theta^\star)\mathbf{J_c}(\theta^\star)^\top\right)^{-1}\mathbf{J_c}(\theta^\star)\nabla_\theta\mathcal{L}(\theta^\star)$. This gives Lagrangian stationarity. $\square$

# F GEOMETRIC INTERPRETATION OF THE NULL-SPACE UPDATE

The explicit null space update rule, as stated above, is given by:

$$\theta_{k+1} = \theta_k - \big[\eta\left(\mathbb{I} - \mathbf{J_c}^\top(\mathbf{J_c}\mathbf{J_c}^\top)^{-1}\mathbf{J_c}\right)\nabla\mathcal{L}(\theta_t) + \gamma\mathbf{J_c}^\top(\mathbf{J_c}\mathbf{J_c}^\top)^{-1}\mathbf{c}(\theta_t)\big] \tag{44}$$

This section provides a geometric perspective to build intuition for the update's components and behavior, formalizing it using concepts from differential geometry.

At any parameter iterate $\theta_t$, the update $\Delta\theta_t$ from Equation 44 can be understood as performing two simultaneous updates related to the local geometry defined by the constraint Jacobian $\mathbf{J_c}(\theta_t)$:

1. **Loss minimization on the tangent hyperplane:** The term $-\eta(\mathbb{I} - \mathbf{J_c}^\top(\mathbf{J_c}\mathbf{J_c}^\top)^{-1}\mathbf{J_c})\nabla\mathcal{L}(\theta_t)$ projects the negative loss gradient onto the null-space of $\mathbf{J_c}(\theta_t)$. This null-space, $\ker(\mathbf{J_c}(\theta_t))$, is the tangent space to the constraint manifold $\mathbf{c}(\theta) = \mathbf{c}(\theta_\mathbf{t})$ at $\theta_t$. This step aims to decrease the loss $\mathcal{L}$ by moving along directions where the linearized constraint value does not change.

2. **Constraint satisfaction step:** The term $\gamma\mathbf{J_c}^\top(\mathbf{J_c}\mathbf{J_c}^\top)^{-1}\mathbf{c}(\theta_t)$ takes a Gauss–Newton step towards satisfying the constraints. This direction lies in the row space of $\mathbf{J_c}(\theta_t)$, $\mathrm{im}(\mathbf{J_c}(\theta_t)^\top)$, which is orthogonal to $\ker(\mathbf{J_c}(\theta_t))$.

These components ensure that the optimization process iteratively reduces the loss while driving the parameters towards the feasible set where $\mathbf{c}(\theta) = \mathbf{0}$.

We can formalize this intuition in the language of Riemannian geometry. For a given parameter $\theta \in \mathbb{R}^D$ and a continuously differentiable constraint function $\mathbf{c}$, we can define two relevant manifolds embedded in $\mathbb{R}^D$. Let $\mathcal{M}_\theta = \{\theta' \in \mathbb{R}^D \text{ such that } \mathbf{c}(\theta') = \mathbf{c}(\theta)\}$ be the kernel manifold where the constraint value is constant and equal to $\mathbf{c}(\theta)$. Its tangent space $T_\theta\mathcal{M}_\theta$ represents directions where the constraint doesn't change locally. Let $\mathcal{N}_\theta$, the image manifold, be a local manifold transversal to $\mathcal{M}_\theta$ at $\theta$, representing directions where the constraint value necessarily changes. Existence of these manifolds is stated and proved formally below:

**Theorem 4.** *For any parameter $\theta$, suppose the set of parameters that have the same constraint value as $\theta$ is denoted by $\mathcal{M}_\theta = \{\theta' \in \mathbb{R}^D \mid \mathbf{c}(\theta') = \mathbf{c}(\theta)\}$. Assuming $\mathbf{J_c}(\theta)$ has full rank, this set is locally a smooth manifold embedded in $\mathbb{R}^D$. Furthermore, there exists a local manifold $\mathcal{N}_\theta$ through $\theta$ such that $\mathbb{R}^D$ can be locally viewed as a product space involving these manifolds, and their tangent spaces $T_\theta\mathcal{M}_\theta$ and $T_\theta\mathcal{N}_\theta$ are orthogonal at $\theta$.*

*Proof.* The existence of these manifolds follow from the preimage theorem. It can be proved as follows: The constraint differential $\mathbf{J_c}(\theta) \in \mathbb{R}^{O \times D}$ is assumed to be full rank. Each column of the $\mathbf{J_c}$ corresponds to the gradient of each constraint

$$\mathbf{J_c}(\theta) = \left(\frac{\partial c_i}{\partial \theta_j}\right)_{i=1,\ldots,O; j=1,\ldots,D} \tag{45}$$

We assume that the differential operator is full-rank, hence surjective, and $D > O$. Consequently, we can reorder the matrix columns to ensure that the first $O$ columns are linearly independent. Then the $O \times O$ matrix below (with reordered columns):

$$R = \left(\frac{\partial c_i}{\partial \theta_j}\right)_{i=1,\ldots,O; j=1,\ldots,O} \tag{46}$$

is invertible. Consider the map

$$\alpha(\theta_1, \ldots, \theta_D) = \left(\frac{\partial c_1}{\partial \theta}, \ldots, \frac{\partial c_O}{\partial \theta}, \theta_{O+1}, \ldots, \theta_D\right) \tag{47}$$

Then we obtain that the Jacobian of $\alpha$, which is:

$$\mathbf{J}_\alpha(\theta) = \begin{pmatrix} R & * \\ 0 & \mathbb{I} \end{pmatrix}. \tag{48}$$

This matrix is invertible. Hence, by the inverse function theorem, $\alpha$ is a local diffeomorphism.

Finally, define

$$\mathcal{M}_\theta = \left\{\alpha^{-1}(\underbrace{\mathbf{c}_1(\theta), \ldots, \mathbf{c}_O(\theta)}_{O \text{ constraint values}}, p_1, \ldots, p_{D-O}) \quad \text{for } p \in \mathbb{R}^{D-O}\right\} \subseteq \mathbb{R}^D, \tag{49}$$

and similarly

$$\mathcal{N}_\theta = \left\{ \alpha^{-1}(p_1, \ldots, p_O, \underbrace{0, \ldots, 0}_{D-O}) \quad \text{for } p \in \mathbb{R}^O \right\} \subseteq \mathbb{R}^D. \tag{50}$$

These the two restrictions of $\alpha$ are slice charts of $\mathcal{M}_\theta$ and $\mathcal{N}_\theta$, respectively, proving that they are embedded manifolds in $\mathbb{R}^D$. $\qquad\square$

We can endow these two manifolds with Riemannian metrics. The kernel manifold $\mathcal{M}_\theta$ inherits the Euclidean metric, i.e., its metric $\mathbf{g}^\perp$ restricted to the tangent space $T_\theta \mathcal{M}_\theta$. The image manifold $\mathcal{N}_\theta$ can be endowed with a metric $\mathbf{g}$ derived by pulling back the Euclidean metric from the constraint output space, such that distances correspond to changes in the constraint value. Hence we get $g = \mathbf{J}_\mathbf{c}^\top \mathbf{J}_\mathbf{c}$, restricted to the tangent space $T_\theta \mathcal{N}_\theta$. Also note that even though $\mathbf{J}_\mathbf{c}^\top \mathbf{J}_\mathbf{c}$ is a low-rank matrix, $g$ is not a pseudo-metric but a proper Riemannian metric because the tangent space of the image manifold excludes directions that live in the null space of the Jacobian and hence of the metric.

We can now interpret the update in Equation 44 (replicated below as Equation 51a) as approximating a Riemannian gradient descent step across these two manifolds. We minimize the loss $\mathcal{L}$ on $\mathcal{M}_\theta$ and the squared constraint norm $||\mathbf{c}(\theta)||^2$ on $\mathcal{N}_\theta$. If we approximate the retractions with the orthogonal projection onto the tangent space, as is standard in the literature (Boumal, 2023), then the Riemannian gradient descent steps on these two manifolds are given by:

$$\theta_{t+1} - \theta_t = -\eta \mathcal{R}_{T_{\theta_t}\mathcal{M}_{\theta_t}}(\mathbf{g}^\perp)^{-1}\nabla\mathcal{L}(\theta_t) - \gamma \mathcal{R}_{T_{\theta_t}\mathcal{N}_{\theta_t}}(-(\mathbf{g})^{-1}\nabla||c(\theta_t)||^2) \tag{51a}$$

$$\approx -\eta \left( \text{proj}_{T_{\theta_t}\mathcal{M}_{\theta_t}}((\mathbf{g}^\perp)^{-1}\nabla\mathcal{L}(\theta_t)) \right) - \gamma \left( \text{proj}_{T_{\theta_t}\mathcal{N}_{\theta_t}}(-(\mathbf{g})^{-1}\nabla||c(\theta_t)||^2) \right) \tag{51b}$$

$$= -\eta \left( (\mathbb{I} - \mathbf{J}_\mathbf{c}^\top(\mathbf{J}_\mathbf{c}\mathbf{J}_\mathbf{c}^\top)^{-1}\mathbf{J}_\mathbf{c})\nabla\mathcal{L}(\theta_t) \right) - \gamma \left( \mathbf{J}_\mathbf{c}^\top(\mathbf{J}_\mathbf{c}\mathbf{J}_\mathbf{c}^\top)^{-1}\mathbf{c}(\theta_t) \right) \tag{51c}$$

This is exactly the null-space update. We can see that the null space method approximates Riemannian gradient descent concurrently on these two manifolds.

# G LOW-RANK LEAST-SQUARES

In the proof of Theorem 1, we always assume that the matrix $\mathbf{A}$ is a full-rank matrix. In this section, we will analyze the gradient computations when $\mathbf{A}$ is a low-rank matrix. However, for differentiability, we still need to assume constant rank in a neighbourhood.

**Damped least-squares:** When the matrix $\mathbf{A}$ is low-rank, perhaps the most important practical case is damped least-squares. This is because regularization is a common way of dealing with ill-posed problems, which corresponds to damped least-squares. Damped least-squares, unlike the cases above, have a unique solution, and their gradients can be derived with only a slight modification to the proof of Theorem 1. We follow the derivation for the tall case, applying the adjoint method to the same constraint

$$(\mathbf{A}^\top\mathbf{A} + \lambda^2\mathbb{I})\mathbf{x} = \mathbf{A}^\top\mathbf{b} \tag{52}$$

This is well defined, even if $\mathbf{A}$ is low-rank, due to the regularization term. Following the exact steps (differentiating this constraint and introducing an adjoint variable $\xi$), we obtain

$$\nabla_\mathbf{b}\mu = \mathbf{A}(\mathbf{A}^\top\mathbf{A} + \lambda^2\mathbb{I})^{-1}\nabla_\mathbf{x}\mu = \texttt{LstSq}(\mathbf{A}, \nabla_\mathbf{x}\mu, \lambda) \tag{53a}$$

$$\xi = -(\mathbf{A}^\top\mathbf{A} + \lambda^2\mathbb{I})^{-1}\nabla_\mathbf{x}\mu \tag{53b}$$

However, unlike the full-rank case, the equation $\xi = -(\mathbf{A}^\top\mathbf{A} + \lambda^2\mathbb{I})^{-1}\nabla_\mathbf{x} = -(\mathbf{A}^\top\mathbf{A})^{-1}\mathbf{A}^\top\nabla_\mathbf{b}\mu$ is not well-defined because $\mathbf{A}^\top\mathbf{A}$ is not invertible. Thus $\xi$ can't be reduced to the same least squares call:

$\mathtt{LstSq}(\mathbf{A}^{\top}, \nabla_{\mathbf{b}}\mu, 0)$. However, with only a slight modification, we can still rewrite $\xi$ as a different least-squares call. Note that

$$(\mathbf{A}^{\top}\mathbf{A} + \lambda^2\mathbb{I})^{-1}\nabla_{\mathbf{x}}\mu = (\mathbf{A}^{\top}\mathbf{A} + \lambda^2\mathbb{I})^{-1}(\mathbb{I} - \mathrm{Proj}_{\mathrm{null}(\mathbf{A})})\nabla_{\mathbf{x}}\mu + (\mathbf{A}^{\top}\mathbf{A} + \lambda^2\mathbb{I})^{-1}\mathrm{Proj}_{\mathrm{null}(\mathbf{A})}\nabla_{\mathbf{x}}\mu \quad (54)$$

$$= \mathtt{LstSq}(\mathbf{A}^{\top}, \nabla_{\mathbf{b}}\mu, 0) + \frac{1}{\lambda^2}(\nabla_{\mathbf{x}}\mu - \mathtt{LstSq}(\mathbf{A}, \mathbf{A}\nabla_{\mathbf{x}}\mu, 0)) \quad (55)$$

This is because the first term finds the minimum norm solution in the range of ($\mathbf{A}^{\top}$, which is the complement of the null-space of $\mathbf{A}$. This corresponds exactly to what the least-squares call computes, the second term lives in the null-space of $\mathbf{A}$, thus it is simply a scaled projection of the rhs ($\nabla_{\mathbf{x}}\mu$) into the null-space of $\mathbf{A}$. This gives us:

$$\xi = \mathtt{LstSq}(\mathbf{A}^{\top}, \nabla_{\mathbf{b}}\mu, 0) + \frac{1}{\lambda^2}(\nabla_{\mathbf{x}}\mu - \mathtt{LstSq}(\mathbf{A}, \mathbf{A}\nabla_{\mathbf{x}}\mu, 0)) \quad (56)$$

With this slightly modified expression of $\xi$, we can proceed with the rest of the derivation, and all the other expressions are the same.

**Undamped tall least-squares:** The undamped tall least squares does not have a unique solution. Notice that if $\mathbf{x}^{\star} = \arg\min_{\mathbf{x}} \|\mathbf{A}(\theta)\mathbf{x} - \mathbf{b}\|^2$ then for any $\mathbf{x}_{\mathrm{ker}} \in \mathrm{null}(\mathbf{A})$, we have that $\|\mathbf{A}(\theta)(\mathbf{x}^{\star} + \mathbf{x}_{\mathrm{ker}}) - \mathbf{b}\|^2 = \|\mathbf{A}(\theta)\mathbf{x}^{\star} - \mathbf{b}\|^2$. Thus $\mathbf{x}^{\star} + \mathbf{x}_{\mathrm{ker}}$ is a solution to the least-squares problem. The $\mathtt{LstSq}$ is then a multi-valued function, and to define gradients, we need additional constraints to select a specific branch of this function.

**Undamped wide least-squares:** The undamped wide case also has a unique solution because it favors the minimum-norm solution by definition Equation 2. Thus, it always sets any null-space component to $\mathbf{0}$. We leave this for future work.

## H    DETAILS ON EXPERIMENTS IN SECTION 3

This section provides detailed information regarding the experimental setups for the results presented in the main paper. Our implementation is developed in JAX (Bradbury et al., 2018), using Optax (DeepMind et al., 2020) for optimization. The code for all experiments is available at [redacted]. The table below provides concrete examples of constrained optimization in deep learning with relevant references

Table 6: Examples for imposing structure via constrained optimization.

| Structure | Constraint $\mathbf{c}_{\psi}(\theta)$ | Key references |
|---|---|---|
| $G$-Equivariance | $T_g(f(\mathbf{x};\theta)) - f(T'_g(\mathbf{x});\theta)$ | Cohen & Welling (2016); Finzi et al. (2021) |
| $G$-Invariance | $f(T_g(\mathbf{x});\theta) - f(\mathbf{x};\theta)$ | Puny et al. (2022) |
| $\ell_0$-Sparsity | $\frac{\|\theta\|_0}{D} - s_{\mathrm{target}}$ | Louizos et al. (2018); Gallego-Posada et al. (2022) |
| Adversarial robustness | $\mathbb{E}_{\mathbf{x}_{\mathrm{clean}},y}[l(f_\theta(x_{\mathrm{clean}}), y)] \leq \delta$ | Robey et al. (2021) |
| Conservativeness | $\mathbb{E}_{\mathbf{p}(\mathbf{x})}[\|\frac{\partial s}{\partial x} - \frac{\partial s}{\partial x}^T\|_F^2]$ | Chao et al. (2023) |

Unless otherwise specified, all deep-learning experiments were conducted on an NVIDIA H100 GPU, and the others on the CPU of a consumer-level laptop. For iterative solvers like LSMR, a tolerance of $10^{-6}$ was used by default. For results reporting mean and standard deviation, experiments were repeated using 3 different random seeds, covering aspects like network initialization and data shuffling.

Table 7: Hyperparameter configurations for constrained optimization experiments.

| Experiment | Model | Optimizer | Weight $\gamma$ | Batch Size | Epochs |
|---|---|---|---|---|---|
| $C_4$ Equivariance | LeNet (FMNIST) | Adam | $10^{-4}$ | 128 | 100 |
| $O(3)$ Equivariance | MLP (Particles) | Adam | 0.5 | 128 | 100 |
| $l_0$ Sparsity | | | | | |
| | ResNet-18 (CIFAR) | Adam | $10^{-4}$ | 128 | 300 |
| | ResNet-18 (SVHN) | Adam | $10^{-4}$ | 128 | 300 |
| | Pre-trained SWIN-S (ImageNet) | SGD | 0.01 | 128 | 10 |
| Conservativeness | MLP (Synthetic 2D) | Adam | 500 | 5000 | $10^5$ |

Section 2.3 applies the null-space method to various constrained optimization problems from the literature. Below, we detail how we implement the constraints for each experiment and report any relevant hyperparameters in Table 7. The only hyperparameter specific to our method is the constraint weight $\gamma$. This weight refers to the constant multiplier of the constraint term in the null-space update. While the convergence is robust to the choice of $\gamma$, this weight $\gamma$ affects the rate of convergence of the constraint.

### H.1   EQUIVARIANCE

$C_4$ **Rotational equivariance on FMNIST**   We enforced $C_4$ rotational equivariance on a LeNet model trained on FMNIST. The constraint function is given by $\mathbf{c}(\theta) = f(Rx; \theta) - Rf(x; \theta)$, for all $x$ and $R \in C_4$, where the output of $f$ is the final output of the convolutional layers of the neural network. Concretely, the constraint measures the norm of the difference between the filters of rotated images and rotated filters of an image, for all the images in a mini-batch and all the rotations in $C_4$. Hence, the constraint output dimension is $B \times 4$, which demonstrates the ability to handle multiple constraints with ease.

The baseline model was trained using Adam with a learning rate of $10^{-3}$ for 100 epochs and a batch size of 128. For the null-space method, the same optimizer was chained with our null-space projection using $\gamma = 10^{-4}$. The constraint violation metric was the mean squared norm $||f(Rx; \theta) - Rf(x; \theta)||^2$ averaged over the test set and all four $C_4$ rotations and test data.

$O(3)$ **Equivariance for particle systems**   This experiment aims to predict the moment of inertia for particle systems, following the task setup by Finzi et al. (2021). Following Finzi et al. (2021), an MLP with three hidden layers of 384 units each and ReLU activations was trained on a synthetic dataset with 5000 data points, each representing a system of particles and their targets corresponding to their respective moments of inertia. Unlike $C_4$, $O(3)$ is not a discrete group. Hence, it is not possible to sample all the group elements. So to enforce the constraints, we sample random matrices from $O(3)$ and the equivariance constraint is given by $f(Rx; \theta) - R^\top f(x; \theta)R = 0$ (Finzi et al., 2021). We average over each mini-batch and end up with a measurement of constraint-violation, which is a $3 \times 3$ inertia tensor. Hence, we have a nine-dimensional constraint.

The baseline uses $O(3)$ data augmentation, where each input particle system was augmented with random $O(3)$ rotations, with corresponding transformations applied to the target tensor. Both the null-space method and baseline uses an Adam optimizer with a learning rate $10^{-3}$ with null space projections with a batch size of 128. Constraint violation was measured as $||f(Rx; \theta) - R^\top f(x; \theta)R||_F^2$, averaged over the test set and random $O(3)$ transformations.

### H.1.1 Enforcing $\ell_0$ sparsity

To enforce $\ell_0$ sparsity, we used learnable stochastic masks $m_i \sim \text{Bernoulli}(p_i)$ for weights $\theta_i = \tilde{\theta}_i m_i$, with the straight-through estimator for gradients of mask probabilities $p$ (Yin et al., 2019). The constraint $c(p) = \frac{1}{N_p} \sum_{i=1}^{N_p} p_i - s_{\text{target}} = 0$ was applied to the expected proportion of active weights. This is a scalar constraint.

**ResNet-18 on CIFAR-10 and SVHN** A standard ResNet-18 architecture is trained on CIFAR-10 and SVHN. Sparsity was targeted at $s_{\text{target}} = 0.1$ (90% sparsity) and applied to [e.g., all convolutional and fully connected layer weights, excluding biases and batch normalization parameters]. The baseline was one-shot magnitude pruning, where the model was trained to convergence, then pruned, and fine-tuned for 20 epochs with a learning rate of $10^{-4}$. For the null-space method, model weights $\theta$ and mask probabilities $p$ were optimized using Adam with learning rates $10^{-3}$ for 300 epochs with batch size 128, with 50 epochs of warm-up, i.e, standard training without any projections. Standard data augmentation for CIFAR-10/SVHN was used (random crops and horizontal flips).

**SWIN-S on ImageNet** A pretrained SWIN-S (Small) transformer was trained on ImageNet with a target sparsity $s_{target} = 0.5$ (50% sparsity). Both methods use an SGD optimizer with a linear one-cycle learning rate schedule and a peak learning rate of $0.1$. The null space method was trained for 10 epochs, with a batch size of 128 and standard ImageNet augmentations.

### H.1.2 Conservativeness of score-based generative models

For score-based generative models, we enforced the conservativeness constraint $c(\theta) = ||J(s) - J(s)^\top||_F^2 = 0$, where $s(x; \theta)$ is the score network and $J(s)$ its Jacobian with respect to $x$. The score network $s(x; \theta)$ was an MLP with Swish activations. The Jacobian $J(s)$ was computed using JAX's automatic differentiation tools per sample, and the constraint was averaged over mini-batches. In this experiment, we attempt to reproduce the setup of Chao et al. (2023) exactly. For additional details on learning rate, batch size, evaluation metrics, and more, refer to Chao et al. (2023).

