# OpenReview forum: "Matrix-Free Least Squares Solvers: Values, Gradients, and What to Do With Them"
_ICLR.cc/2026/Conference — Submitted to ICLR 2026_

### Official Review · Reviewer_knUw · 2025-11-01

**Soundness:** 3
**Presentation:** 2
**Contribution:** 3
**Rating:** 6
**Confidence:** 4

**Summary:**

The authors propose an efficient VJP method to backpropagate through the solution of a least square operation, thus
enabling it as a differentiable operation in any ML workflow.

**Strengths:**

* Tacking an important linear algebra primitive such as LstSq is potentially of high impact.
* Framing the LstSq operation as a layer is novel and though provoking.
* The VJP implementation that avoid backpropagating through iterative algorithms is theoretically well motivated,
  efficiently implemented and functional.

**Weaknesses:**

* Some of the applications do not show the full potential of the method. In particular the Gaussian Process example  is
  small in terms of the features of the data, the number of hyperparameters and the use of random Fourier features
  (which is not the common practice)
* The method just applies to full rank matrices.

**Questions:**

* Line 093: What does it mean to require "twice the precision". You mean potentially twice the number of iterations?
  Precision could also suggest going from float32 to float64 or float128 but I doubt that you are referring to this.
  If you are referring to numerical precision. What precision do you use in your experiments? The standard float32 in
  deep learning?
* Are you familiar with [1]? The authors also get transposition automatically as a VJP and offer a series of matrix
  operations with differentiation both in JAX and PyTorch. I believe this work should be discussed in your related work
  section.

[1] Potapczynski et al 2023. CoLA: Exploiting Compositional Structure for Automatic and Efficient Numerical Linear Algebra

---

> ### Author Response · Authors · 2025-11-25
>
> Thank you for your thoughtful review. We address your concerns below.
>
> > The method just applies to full rank matrices
>
> Thank you for highlighting this crucial point;  several reviewers raised it. We have now extended Theorem 1 to also include the gradients for the rank-deficient least-squares problem with non-zero damping. We also include a discussion of the zero-damping case. Please have a look at Appendix G in the updated PDF.
>
> >  Some of the applications do not show the full potential of the method. In particular the Gaussian Process example is small in terms of the features of the data, the number of hyperparameters and the use of random Fourier features (which is not the common practice)
>
> We use feature-based Gaussian processes because they have a natural least-squares formulation, and among feature-based Gaussian processes, random Fourier features are the most typical.  General Gaussian processes are also admissible as long as they have tractable square roots of covariance matrices. Also, we would like to highlight that the Gaussian process experiment is only one of our application-level contributions, alongside several others. To demonstrate scalability, we have different experiments on Imagenet with a Vision Transformer.
>
> > Line 093: What does it mean to require "twice the precision". You mean potentially twice the number of iterations? Precision could also suggest going from float32 to float64 or float128 but I doubt that you are referring to this. If you are referring to numerical precision. What precision do you use in your experiments? The standard float32 in deep learning?
>
> By precision, we mean exactly “going from float32 to float64”. The reason we say this is because normal equations form the system $AA^\top$, effectively squaring the conditioning of the system, whereas orthogonal transformations strictly deal with $A$. So to achieve the same robustness with these two methods, one would have to increase the machine precision for the normal equation. All experiments are done in float32.
>
> > Are you familiar with [1]? The authors also get transposition automatically as a VJP and offer a series of matrix operations with differentiation both in JAX and PyTorch. I believe this work should be discussed in your related work section.
>
> Thank you for pointing out this oversight on our part. We are familiar with [1] and completely agree that it is related to our work. We have expanded the related works section to include a discussion of [1].
>
> Thank you for your constructive comments. We hope that we have addressed your concerns and look forward to the rest of the discussion.

---

### Official Review · Reviewer_njXR · 2025-11-01

**Soundness:** 3
**Presentation:** 3
**Contribution:** 3
**Rating:** 6
**Confidence:** 4

**Summary:**

- The paper treats the least-squares solver as an operator and derives its vector–Jacobian product when used as the input to a scalar function.
- Building on the null-space method, the authors reformulate constrained optimization problems into unconstrained forms that incorporate the least-squares operator.
- The authors validate experimentally the two procedures

**Strengths:**

- The paper is well-written with concise explanations and helpful supplementary material in the appendix.
- The proposed VJP procedure demonstrates strong computational efficiency, as supported by timing comparisons.
- The introduced framework and accompanying library provide a practical and flexible means to incorporate constraints optimization setups.

**Weaknesses:**

**Nuancing statements**
In Section 2.1, matrix-free methods are presented as ideal, yet it must be must highlighted that they also require access to a procedure to evaluate $A^\top u$.
When such access is unavailable, matrix-free approaches lose their advantage and can become as expensive as direct linear-system methods.
Similarly, Equation (4), which deduce $A^\top u$ via a vjp computation, should be tempered: vjp can introduce non-negligible computational overhead; see [2, Table 7.1].

**Ambiguity between Sections 2.2 and 2.3**
The connection between Sections 2.2 and 2.3 is unclear. Section 2.3, on constrained optimization via least squares, appears largely independent of the vjp derivations in 2.2. Since the results of 2.2 are not required for the developments in 2.3, the transition feels abrupt and breaks the logical flow of the paper. The authors should clarify how these sections relate conceptually or reorganize the exposition to improve coherence.

**Positioning within the literature and experimental context**
The discussion of prior work on vjp computation is limited. The manuscript should better position its contribution relative to existing approaches, particularly those addressing implicit differentiation (e.g., [1]) and direct derivations of the VJP for the least-squares operation (e.g., [3]).
In particular, it is not clear what are the pros and cons of the proposed derivation relative to that.
This incompleteness carries into the experiments, where the proposed method is evaluated against few baselines, for instance two bases for runtime comparison


**Typos/minor issues**

* Line 94: missing $\theta$ in front of $A$.
* Lines 110–111: replace "By reducing the likelihood of errors" ---> “By reducing sources of errors.”
* Line 219: define the abbreviation PGD (Projected Gradient Descent) as it is used right after in Line 221-222
* Section 2.3: explicitly state that the number of constraints is smaller than the parameter dimension, as assumed in Theorem 3


---

.. [1] Blondel, Mathieu, et al. "Efficient and modular implicit differentiation." Advances in neural information processing systems 35 (2022): 5230-5242.

.. [2] Blondel, Mathieu, and Vincent Roulet. "The elements of differentiable programming." arXiv preprint arXiv:2403.14606 (2024).

.. [3] Wan, Zhou-Quan, and Shi-Xin Zhang. "Automatic differentiation for complex valued SVD." arXiv preprint arXiv:1909.02659 (2019).

**Questions:**

- Line 243-245: in the experiment design, can you clarify "increasing number of rows and columns"?
- Full-rank assumption: The method relies on the matrix being full-rank. How realistic is this assumption in practical applications, especially when dealing with ill-conditioned or rank-deficient systems, e.g. in inverse problems?

---

> ### Author Response · Authors · 2025-11-25
>
> We thank the reviewer for their helpful comments, we are grateful for their thorough engagement with our work, and we address their points below.
>
> > **Nuancing statements**
>
> These are fair points. We agree that if efficient custom vector matrix products are available, they should be used. We also agree that there are cases where direct methods are preferable over iterative methods, especially in cases where the matrix is small and ill-conditioned. In instances where vector matrix products are not available as easily, we offer an automatic way of determining them via reverse-mode autodiff. Regarding your reference, Table 8.1 in [2] describes the cost of populating Jacobian matrices using reverse-mode differentiation, whereas our Equation 4 does a single backward pass through a function that is known to be linear (knowing linearity makes reverse-mode autodiff very efficient, c.f. Theorem 7.1 in “You Only Linearize Once” by Radul et al.). In any case, we observed in our experiments that this automatic vector matrix product has never been a bottleneck. Thank you for bringing this up. We have updated the PDF to address your concerns.
>
> > **Ambiguity between Sections 2.2 and 2.3**
>
> Thank you for raising this point. We organize the methods section to match the structure of the title because we believe that this is the clearest way of highlighting all our contributions. In the title, we talk about values, gradients, and what to do with them. This corresponds to sections 2.1 (values of least squares), 2.2 (gradients of least squares), and 2.3 (applications of least squares). So you are correct that these sections are technically independent of each other, but we want to present all contributions in Section 2.
>
> > **Positioning within the literature and experimental context**
>
>  We have now expanded the discussion of the related works in Section 2.2. With regards to your questions about highlighting the differences:
>
> [1] implements software for implicit differentiation of various optimality conditions for general problems. [3] derives adjoints for SVD, which is one of many ways to solve least-squares problems. However, [3] does not discuss least-squares.
>
> Our work is similar to [1] in the sense that we both use implicit differentiation to obtain efficient gradients of functions. Our work is similar to [3] because we derive gradients of least-squares, and a different way of computing these gradients can be arrived at as a corollary of Eq. 35 in [3]. However, there are also significant differences. Our work differs from [1] by being more specialised and focusing on least-squares problems, and this specialisation gives us quite a significant advantage in computational efficiency (essentially by avoiding general-purpose linear system solvers like CG or GMRES; Appendix A explains why they are inferior to LSMR for least-squares problems). In a similar vein, the SVD derivatives could be used to obtain least-squares gradients, but LSMR is more efficient than SVD (Table 1).
>
> > Full-rank assumption: The method relies on the matrix being full-rank. How realistic is this assumption in practical applications, especially when dealing with ill-conditioned or rank-deficient systems, e.g. in inverse problems?
>
> Thank you for highlighting this crucial point;  several reviewers raised it. We have now extended Theorem 1 to also include the gradients for the rank-deficient least-squares problem with non-zero damping. We also include a discussion of the zero-damping case. Please have a look at Appendix G of the updated PDF. Specifically for inverse problems, rank-deficient systems are typically addressed by regularization, and the new extension of Theorem 1 covers this case.
>
> > Line 243-245: in the experiment design, can you clarify "increasing number of rows and columns"?”
>
> By increasing the number of rows and columns, we run multiple experiments, each with a different-sized matrix.
>
> > Typos/minor issues
>
> Thank you very much for pointing these out! We have updated the paper to fix these errors.
>
> Thanks again for your thoughtful comments. We believe they have helped us improve the paper, especially the extension to the rank-deficient systems. We look forward to the discussion!

---

### Official Review · Reviewer_vaDD · 2025-11-01

**Soundness:** 3
**Presentation:** 3
**Contribution:** 3
**Rating:** 6
**Confidence:** 4

**Summary:**

This paper introduces a differentiable, matrix-free least-squares operator that turns classical solvers like LSMR into drop-in neural network layers with custom gradients. The authors derive reverse-mode derivatives for tall and wide matrices (Theorem 1) and provide a JAX implementation that automatically constructs $A^\top$ via vector-Jacobian products. They further reinterpret the null-space method for constrained optimization as a least-squares projection, enabling the use of standard Optax optimizers for constraint enforcement. Applications include Gaussian process calibration via differentiable least squares, symmetry and sparsity constraints in deep nets, and conservative score-based generative models. Empirically, custom gradients yield 5–10× runtime improvement over solver unrolling, and the null-space formulation produces strong results on large models with minimal code overhead.

**Strengths:**

The paper has the following strengths:

- **Originality:** Theorem 1 extends beyond prior work by Golub & Pereyra (1973) and Krämer et al. (2024) by providing reverse-mode derivatives for adaptive least-squares solvers with regularization terms, handling both tall and wide cases in a unified framework. The reformulation of Yamashita (1980)'s null-space algorithm as a least-squares projection is particularly creative, enabling practical large-scale constrained optimization on models with up to 50 million parameters.

- **Quality:** The technical quality is good, with rigorous mathematical foundations throughout. The proofs in Appendix C are complete and correct, and the distinction between tall and wide matrices with their limiting behaviors is thoroughly analyzed. The experimental validation is comprehensive, spanning five diverse applications that demonstrate broad applicability.

- **Clarity:** The paper writing is smooth, which progresses logically from fundamentals to gradients to applications, and the appendices effectively organize background material and detailed proofs.

- **Significance:** The work addresses a significant gap in the machine learning toolkit by making numerical least-squares solvers more accessible in differentiable programming frameworks. The application of Gaussian process calibration is particularly noteworthy, as directly optimizing predictive fit is simpler and more scalable than marginal-likelihood optimization, yet surprisingly underexplored in the literature.

**Weaknesses:**

- **Full Rank Assumption**: The most critical limitation is the full-rank assumption that pervades the entire paper. The paper does not guide diagnosing rank deficiency, choosing regularization λ to stabilize near-singular cases, or understanding what happens numerically when the assumption is violated. Even when matrices are technically full rank, severe ill-conditioning can produce similar numerical failures. Moreover, no experiments report the numerical rank or condition number of $J_c$, leaving it unclear whether the demonstrated applications actually satisfy the full-rank requirement or whether the method is more robust than the theory suggests.

- **Practical Applicability**: The paper provides no guidance on how practitioners should diagnose rank deficiency before applying the method, what happens numerically when assumptions are violated, when or how to choose regularization λ to stabilize near-singular cases, or whether the technique degrades gracefully or fails catastrophically. This limitation is not mentioned in the abstract and is only briefly acknowledged in the conclusion. Given that the title promises "Matrix-Free Least Squares Solvers" for "modern machine learning," the full-rank restriction substantially narrows the scope. The paper should prominently scope the work to full-rank problems with a clear justification of their practical relevance compared to non-full-rank matrices.

- **Insufficient Numerical Analysis**: The paper uses a tolerance of $10^{-6}$ for LSMR but does not investigate how backward-pass tolerances affect gradient accuracy. While Example 2 shows that LSMR handles condition numbers around $10^7$ well for the forward pass, the backward pass requires solving two additional least-squares problems, as stated in Theorem 1. No experiments report condition numbers, ranks, or numerical diagnostics for the constraint Jacobians.

- **Limited discussion on preconditioning**: Preconditioning is essential for practical large-scale problems, yet readers have no guidance on how custom gradients interact with preconditioners or whether users can supply preconditioners to the LSMR implementation.

**Questions:**

1. Do backward passes require tighter tolerances than forward passes? What is the relationship between solver tolerance and gradient error?
2. Have the authors validated the custom gradients against finite differences or double-precision automatic differentiation? How sensitive are the gradients to LSMR tolerance: should backward passes use tighter tolerances than forward passes?
3. What is the effective rank and conditioning of $J_c$ in the equivariance experiments, and for SWIN-S with 50M parameters? What is the rank and conditioning of the constraint matrix?
4. Based on the response to Q3, can the authors comment on the generality of the proposed matrix-free least square solver?

---

> ### Author Response · Authors · 2025-11-25
>
> We thank the reviewer for their excellent review and thorough engagement with our work.
>
> > **Full Rank Assumption**
>
> Thank you for highlighting this crucial point;  several reviewers raised it. We have now extended Theorem 1 to also include the gradients for the rank-deficient least-squares problem with non-zero damping. We also include a discussion of the zero-damping case. Please have a look at Appendix G of the updated PDF.
> The values of least-squares are unaffected by rank deficiency because LSMR always returns the minimum norm solution even if the least-squares problem does not have a unique solution (for tall, undamped least-squares). In any case, for rank-deficient systems, it is usual to consider the damped/regularized case where the solution is unique.
>
> > The paper does not guide diagnosing rank deficiency
>
> Our implementation of the least-squares solver returns the solution and a set of diagnostic criteria, including the estimated conditioning of A, which helps in diagnosing rank deficiency. We don't report it in the experiments for clarity of presentation, but they are accessible if needed. We don’t observe severe ill-conditioning in any of our experiments.
>
> > Even when matrices are technically full rank, severe ill-conditioning can produce similar numerical failures.
>
> Our implementation of the LstSq operator is an adaptive solver, meaning it runs for as many iterations as necessary to satisfy a stopping criterion, which includes a numerical estimation of the conditioning of A and can be specified when the solver is initialized. Therefore, many of these concerns, while reasonable, are addressed in the solver design. But you are certainly right that severe ill-conditioning requires careful method selection. This is also why we recommend solving least-squares via orthogonal transformation (LSMR) rather than normal equations (CG), as this avoids squaring the conditioning.
>
> > **Choosing a regularization constant**
>
> General guidelines for choosing $\lambda$ would be nice indeed, but this is highly context-dependent. For example, in the case of a Bayesian approach to an ill-posed inverse problem with rank-deficient A, the damping might come from the choice of the prior. Otherwise, we recommend choosing the smallest $\lambda$ that gives sensible results, which can only be discovered empirically.
>
> > **on preconditioning**
>
> Thank you for raising this important point. With regards to values (or solving least-squares problems): We did not find preconditioners to be necessary for any of our experiments, since none of the matrices were sufficiently ill-conditioned. However, in cases where preconditioners are deemed to be necessary, they can easily be incorporated in our implementation of LstSq, since they can be absorbed into the matvecs. The discussion on weighted least-squares in Appendix D outlines how this can be done. With regards to differentiating through preconditioned least-squares problems: The results in Theorem 1 can be used directly to obtain the gradient wrt a preconditioner. Since preconditioner can be absorbed into the matvecs/vecmats, we can treat it as part of the parameter ($\theta$) and compute the parameter gradients.

---

> > ### Author Response · Authors · 2025-11-25
> >
> > > Do backward passes require tighter tolerances than forward passes? What is the relationship between solver tolerance and gradient error?
> >
> > We always use the same tolerances for the forward pass and the backward pass. The gradient error is of the same order as the tolerance for the forward problem. In the table below, we follow a similar setup as in experiment 3.1, where we have a convolution matrix of size 4096. We solve the least-squares problem via LSMR with different tolerances and compute the gradient of a scalar function that depends on the solution (squared norm of the solution). We then compare both the gradients with respect to matrix parameters and the right-hand-side obtained via our method and the ones obtained via backpropagation through the while loop (Figure 3). We obtain the following differences (relative RMSE)
> >
> > | LSMR Tol  | RHS gradient difference    | Matrix gradient difference  |
> > |------------|--------------------------|---------------------------|
> > | 1.0e-01    | 9.65e-03      | 1.22e-02       |
> > | 1.0e-02    | 7.50e-03      | 5.32e-03       |
> > | 1.0e-03    | 4.24e-04      | 2.48e-04       |
> > | 1.0e-04    | 1.80e-05      | 7.19e-06       |
> > | 1.0e-05    | 1.37e-05      | 1.81e-05       |
> > | 1.0e-06    | 4.59e-06   | 4.09e-06          |
> >
> > They show that the gradient differences have a similar magnitude as the LSMR tolerance, which makes sense because the gradient computation is another least-squares call on the same system or the transposed system.
> >
> >
> > > Have the authors validated the custom gradients against finite differences or double-precision automatic differentiation?
> >
> > Yes, we have unit tests that compare against SVD gradients and autodifferentiation gradients. We do not compare against finite differences because the other two are more accurate.
> >
> > > What is the effective rank and conditioning of in the equivariance experiments, and for SWIN-S with 50M parameters? What is the rank and conditioning of the constraint matrix?
> >
> >
> > Using the diagnostics returned by our solver, we compute the conditioning of the Constraint Jacobian on a single batch:
> >
> > Equivariance (ResNet on FMNIST) condition number: 20.87
> >
> > Sparsity(SWIN-S on ImageNet) condition number: 7043.17
> >
> > These results show that the constraint Jacobian is quite well-conditioned.
> >
> > > Based on the response to Q3, can the authors comment on the generality of the proposed matrix-free least square solver?
> >
> >   According to our experiments, the matrix-free least squares solvers are robust in realistic settings in constrained optimization tasks. However, the conditioning and stability of the system in question are likely context-dependent.
> >
> > Thank you again for your detailed review. Engaging with them helped us improve the paper, and we look forward to your response.

---

### Official Review · Reviewer_oBes · 2025-11-02

**Soundness:** 2
**Presentation:** 2
**Contribution:** 1
**Rating:** 2
**Confidence:** 5

**Summary:**

Authors propose a method to differentiate efficiently through the solution of a linear system, e.g., if one defines $x^{\star}(A, b)$ as the solution of $Ax^{\star} = b$, then one can differentiate $x^*$ with respect to $A$ or $b$. Authors have experiments to demonstrate that the proposed gradient computation method is faster then other methods, and show that their method can be useful for Gaussian process calibration and constrained optimization.

**Strengths:**

The topic of differentiating through optimization problem is interesting, especially since the resurgence of implicit differentiation-based approach like "Tiny Recursive Models".

**Weaknesses:**

Weaknesses:
- Clarity:
    - I did not understand the proposed approach, do authors use a different solver than conjugate gradient? and then implicit differentiation?
    - What does the costs in Table 1 represent? To my knowledge, direct methods are $O(n^3)$. Is this the memory cost? Is this the computer cost of "one iteration" of each method?
    - What exactly is the proposed approach? Is it equation 10b?

- Novelty:
    - I do not understand the novelty, especially since authors use an iterative solver: solving a linear system is equivalent to solve an optimization problem (as authors themselves state before equation 1), that is solve using well-known iterative methods. I.e., differentiating through the solution of linear system is equivalent to differentiate through the solution of an optimization problem.
    - I know that authors mentioned [1] in their related work, but could authors extend on the difference between their work and [1]? It would really help me understand the contribution of he paper. What is the difference with approaches like [2]?

- Experiments
    - IMO experiment in 3.1/Figure 3 does not meet top-conference standard requirements, and can be prone to misunderstanding: authors display the wall time as a function of the size of the problem. However, all the displayed methods are iterative, how was the number of iterations of each method was chosen/fixed? If the number of iteration is  not prefixed, what is the stopping criterion? Usually, methods all have different stopping criterion, which make them harder to compare. This details are not mentioned, nor in the main paper, nor in the appendix.  IMO, not mentioning this crucial details make this experiment inconclusive.
    - What are the baseline in Figure 3? I cannot find any mention of "AD (chekcpointed)" in the manuscript. Could authors compare to standard implicit differentiation [2]? with multiple inner solver, like conjugate gradient and/or the suggested one?
    - I do not understand how equation 12 differs from the setting in [2]

[1] Brandon Amos and J Zico Kolter. Optnet: Differentiable optimization as a layer in neural networks

[2] Pedregosa, Fabian. "Hyperparameter optimization with approximate gradient." International conference on machine learning. PMLR, 2016.

**Questions:**

cf weaknesses

---

> ### Author Response · Authors · 2025-11-25
>
> We thank the reviewer for their comments. We will make some general remarks about clarity, novelty, and contributions, and then respond to each point individually:
>
> **Contribution Summary**
>
> The paper is structured to mimic the paper title: we study how to solve least-squares problems ("values"), how to differentiate the solvers ("gradients"), and applications to state-of-the-art machine learning ("what to do with them"):
>
> 1. Values: Section 2.1 reviews the different approaches to solving least-squares problems and their trade-offs (Table 1 & Appendix A). This section is not a novel contribution per se, but we also provide an efficient open-source implementation of least-squares solvers in JAX, which easily fit into deep learning pipelines.
> 2. Gradients: We propose a new method for differentiating through least-squares solvers. We derive and implement custom adjoints of least-squares problems (Theorem 1), which we demonstrate to have clear advantages over naively backpropagating through the solver (Figure 3). This is both a theoretical and practical contribution.
> 3. What to do with them: We demonstrate how to use these values and gradients in machine learning tasks that everyone cares about:
>  -  How to use the values for constrained optimization of neural networks for very general constraints by reformulating the classical null space method as a least-squares problem. This is a methodological contribution.
> - How to use gradients to calibrate Gaussian processes by directly optimizing the posterior fit instead of marginal-likelihood optimization. This is also a methodological contribution.
>
> In summary, this paper makes several contributions, all of which are connected through a study of least-squares solvers in modern machine learning.

---

> > ### Author Response · Authors · 2025-11-25
> >
> > **Response to individual points**
> >
> > > I did not understand the proposed approach, do authors use a different solver than conjugate gradient? and then implicit differentiation?
> >
> > Yes, in the forward pass, we recommend using solvers such as LSMR instead of conjugate gradients (see Table 1 and Figure 5). Briefly, using conjugate gradients squares the conditioning of the system, requiring higher precision. In the backward pass, our method is a form of implicit differentiation specialized to least-squares: rather than unrolling the iterations, we derive a closed-form adjoint system (Theorem 1) and solve it with the same least-squares solver as for the forward pass.
> >
> > > What does the costs in Table 1 represent? To my knowledge, direct methods are $O(n^3)$. Is this the memory cost? Is this the computer cost of "one iteration" of each method?
> >
> > Yes, the costs as stated in Table 1 represent the memory cost. Furthermore, $O(n^3)$ represents the computational complexity of linear solves (with an invertible square matrix). Least squares problems are more general (see equation 2).
> > > What exactly is the proposed approach? Is it equation 10b?
> >
> > Yes, Equation (10b) is one of our contributions. See the summary of contributions above.
> >
> > > I know that authors mentioned [1] in their related work, but could authors extend on the difference between their work and [1]? It would really help me understand the contribution of he paper. What is the difference with approaches like [2]?
> >
> > Regarding novelty, both are indeed closely related to matrix-free least squares problems. However, there are some important differences that we would like to highlight:
> > - **Linear solvers:** Both our method and linear-solver derivatives use implicit differentiation (respectively the adjoint method) to avoid backpropagating through long, data-dependent while loops. However, the key difference is that linear systems $Ax=b$ require $A$ to be square and invertible, whereas least squares problems (our work)
> > $$
> > \|Ax-b\|^2 + \lambda^2 \|x\|^2 \rightarrow \text{min}
> > $$
> > handle non-square or non-invertible matrices. This difference in scope means that existing formulas for linear-system gradients cannot be used for least-squares problems, and our work provides those missing pieces. Fortunately, despite the increased generality of our algorithm, the complexity of each of our custom backward passes is comparable to linear-system derivatives.
> > - **Quadratic programs:** Least-squares problems (our work) are a subset of quadratic programs (considered by [1]), and do not include, for instance, inequality constraints. However, this specialisation leads to significant advantages in computational efficiency: Our work leverages fast, matrix-free least-squares solvers like LSMR and directly computes gradients wrt to the parameters of matrix-vector products. In contrast, [1] must rely on interior point methods and LU decompositions to evaluate gradients, which are substantially more expensive and do not admit matrix-free formulations. Crucially, since [1] works with direct methods, the memory cost makes it infeasible in modern machine learning, where we could be dealing with matrices that have billions of elements. Does this explanation help? If so, we'd be happy to include the above information in a revised version of the manuscript, e.g., at the end of Section 2.2.
> >
> > > IMO experiment in 3.1/Figure 3 ... However, all the displayed methods are iterative; how was the number of iterations of each method was chosen/fixed?
> >
> > > What are the baseline in Figure 3? I cannot find any mention of "AD (chekcpointed)" in the manuscript. Could authors compare to standard implicit differentiation [2]? with multiple inner solver, like conjugate gradient and/or the suggested one?”
> >
> > Figure 3 compares three different ways of computing the backward pass while keeping the forward pass fixed. So while you are correct that the forward pass is iterative, they are run for the same number of iterations / stopping criterion. The only difference is how we backpropagate through the forward pass.
> > Baselines in Figure 3 represent the gradients obtained by backpropagating through the iterative algorithms. Standard JAX while loops do not have reverse-mode derivatives, so we implement LSMR with Equinox’s while loops as a reverse-mode differentiable baseline. Equinox implements two such while loops (either checkpointed or bounded), which are our two baselines. Figure 3 shows that our implementation of LSMR with custom gradients is orders of magnitude faster than autodiff gradients. We have updated the paper to better clarify this. Regarding [2], it uses conjugate gradients, which would be an uninformative baseline because Figure 3 does not compare different solvers but different gradients of the same solver.
> >
> > Thanks again for your review. We hope that our response addresses your concerns adequately. In any case, we look forward to the discussion.

---

### Author Response · Authors · 2025-12-01
**Summary of reviews and rebuttal**

We see that reviewers will no longer be able to reply. Therefore, we would like to briefly summarize the main contributions of our work, the main points raised by the reviewers, and our response.

In this paper, we study

1. how to best solve least-squares problems ("values"); we **contribute** a comparative analysis of different methods and an efficient open source library containing the recomended method (LSMR),
2. how to efficiently differentiate the solvers ("gradients"); we **contribute** a derivation and implementation of custom adjoints of least-squares problems which is magnitudes faster than naive backpropogation (turning least-squares solvers into the equivalent of a neural network layer),
3. applications to state-of-the-art machine learning ("what to do with them"); we **contribute** a method for a plug-and-play constrained optimizer for neural networks for general constraints. The gradients of least squares are used to develop a novel method for calibrating Gaussian processes.

The initial reviews' main criticism (mentioned by Reviewers knUw, njXR, and vaDD) was the full-rank assumption in Theorem 1. This assumption was removed during the rebuttal period with our extension of Theorem 1 (now Appendix G in the updated PDF). We also clarified the other points, e.g. oBes' questions about the relationship of the least-squares solver with linear-system solvers and quadratic programming, and were looking forward to the reviewers' responses.

We want to thank everyone for their engagement with our submission and deeply appreciate their efforts.

---

### Meta-Review · Area_Chair_ye5Z · 2026-01-08

**Summary:**

This paper proposes a matrix‑free least‑squares operator with application to constrained optimization and Gaussian process calibration. The paper is supported by a thorough empirical section and is clearly written, together with a JAX implementation. Reviewers also raise concerns about novelty, the rigor of the experiments, and limitations of the scope of the method; in particular one reviewer would have liked more context for the core contribution relative to the mature literature on differentiating through linear systems. Ultimately, the paper would be much improved with a stronger methodological core and theoretical depth, and can be improved via the contructive comments of the referees.

**Reviewer Concerns:**

There is remaining concern on the novelty compared to many well studied iterative solvers for linear systems. While the experimental section is thorough, it lacks some details on baselines, tolerances/stopping conditions etc, and could benefit from further round of review on these points. Together with some other details like preconditioning and the lack of contextualization, some concerns remain outstanding.

**Reviewer Scores:**

The paper received two borderline positive reviews and a stronger opinion on reject and the content is consistent with the scores

---

### Decision · Program_Chairs · 2026-01-26

Reject